# Autonomous and Sustainable Service Economies: Data-Driven Optimization of Design and Operations through Discovery of Multi-Perspective Parameters

Nala Alahmari [1], Rashid Mehmood [2,*], Ahmed Alzahrani [1], Tan Yigitcanlar [3] and Juan M. Corchado [4,5,6]

[1] Department of Computer Science, Faculty of Computing and Information Technology, King Abdulaziz University, Jeddah 21589, Saudi Arabia; nsaadalahmari0001@stu.kau.edu.sa (N.A.); asalzahrani@kau.edu.sa (A.A.)

[2] High-Performance Computing Center, King Abdulaziz University, Jeddah 21589, Saudi Arabia

[3] City 4.0 Lab, School of Architecture and Built Environment, Queensland University of Technology, Brisbane, QLD 4120, Australia; tan.yigitcanlar@qut.edu.au

[4] BISITE Research Group, University of Salamanca, 37007 Salamanca, Spain; jm@corchado.net

[5] Air Institute, IoT Digital Innovation Hub, 37188 Salamanca, Spain

[6] Department of Electronics, Information and Communication, Faculty of Engineering, Osaka Institute of Technology, Osaka 535-8585, Japan

[*] Correspondence: r.mehmood@gmail.com

**Abstract:** The rise in the service economy has been fueled by breakthroughs in technology, globalization, and evolving consumer patterns. However, this sector faces various challenges, such as issues related to service quality, innovation, efficiency, and sustainability, as well as macro-level challenges such as globalization, geopolitical risks, failures of financial institutions, technological disruptions, climate change, demographic shifts, and regulatory changes. The impacts of these challenges on society and the economy can be both significant and unpredictable, potentially endangering sustainability. Therefore, it is crucial to comprehensively study services and service economies at both holistic and local levels. To this end, the objective of this study is to develop and validate an artificial-intelligence-based methodology to gain a comprehensive understanding of the service sector by identifying key parameters from the academic literature and public opinion. This methodology aims to provide in-depth insights into the creation of smarter, more sustainable services and economies, ultimately contributing to the development of sustainable future societies. A software tool is developed that employs a data-driven approach involving the use of word embeddings, dimensionality reduction, clustering, and word importance. A large dataset comprising 175 K research articles was created from the Scopus database, and after analysis, 29 distinct parameters related to the service sector were identified and grouped into 6 macro-parameters: smart society and infrastructure, digital transformation, service lifecycle management, and others. The analysis of over 112 K tweets collected from Saudi Arabia identified 11 parameters categorized into 2 macro-parameters: private sector services and government services. The software tool was used to generate a knowledge structure, taxonomy, and framework for the service sector, in addition to a detailed literature review based on over 300 research articles. The conclusions highlight the significant theoretical and practical implications of the presented study for autonomous capabilities in systems, which can contribute to the development of sustainable, responsible, and smarter economies and societies.

**Keywords:** service economy; smart society; autonomous services; smart services; sustainable services; deep learning; big data analytics; Natural language Processing (NLP); internet of things (IoT)

## 1. Introduction

The rise in the service economy, driven by technological advancements, globalization, and changes in consumer behavior, has transformed our way of life, our occupations, and social interactions [1,2]. Services such as banking, education, healthcare, tourism, and

entertainment have become the main drivers of economic growth, replacing traditional industries such as manufacturing and agriculture [3–5]. This shift towards a service-based economy has created employment opportunities, increased standards of living, and brought about greater innovation and personalization in service provision [6–9].

### 1.1. Macro and Micro Challenges Facing Service Economies

The services sector has become a crucial part of many modern economies, accounting for approximately 71% of global GDP in 2020, up from 60% in 1990 [10,11]. This growth has been linked to increases in innovation, productivity, and job creation, particularly in the digital age enabled through technologies such as big data, the internet of things (IoT), artificial intelligence (AI), and others [3,12–14]. Service exports have also become an important source of revenue for many countries, with recent studies highlighting their positive impact on economic growth and development [15]. Moreover, the International Labor Organization reports that the services sector is the largest employer in the world, providing jobs to approximately 50% of the global workforce [16,17].

While the service sector has brought unprecedented advancements and comfort to our lives, service economies face a range of challenges, both at the macro and micro levels. At the macro level, globalization has increased competition and put pressure on service providers to innovate and increase efficiency so as to remain competitive [2]. Globalization can also lead to various negative impacts on local service economies, such as economic instability, social unrest, environmental degradation, cultural erosion, and health risks. Rapid globalization can result in economic instability for local businesses and services [18,19]. Additionally, social unrest can arise due to growing income inequality and job insecurity, leading to political tensions and even violence [20]. Globalization can also lead to environmental degradation, as increased production and consumption result in higher carbon emissions and other forms of pollution [21,22]. Furthermore, cultural erosion can occur as a result of the global spread of homogenized services, leading to a loss of local cultural identity [20]. Globalization can also lead to health risks, such as the spread of infectious diseases through increased travel and trade [23].

Geopolitical risks such as trade disputes and political instability can also affect service industries by disrupting global supply chains and causing uncertainty [24]. Recessions can have a significant impact on service economies, with decreased consumer spending leading to reduced demand for services [25,26]. Bank and company defaults and collapses, such as the recent falls of Silicon Valley Bank, Silvergate, Credit Suisse, and Signature Bank, can also have a ripple effect on service industries by reducing consumer confidence and causing supply chain disruptions [27]. Technological disruptions, such as the rise in automation and artificial intelligence, pose both opportunities and challenges for service economies [28,29]. While automation can increase efficiency and reduce costs, it can also lead to job losses and the need to retrain workers [28,30]. Climate change is another major challenge facing service economies, particularly those reliant on tourism and outdoor recreation [31]. Extreme weather events and natural disasters can disrupt service industries, while climate change mitigation efforts can also require significant investments and changes to service provision [32].

Demographic shifts, including aging populations and changing family structures, can also impact service industries [33]. As the population ages, demand for healthcare and other services may increase, while changing family structures can lead to changes in demand for childcare and other services [34,35]. Income inequality is another challenge facing service economies, as it can lead to decreased demand for certain services and create social unrest [20]. Regulatory changes, such as new data privacy regulations and labor laws, can also create challenges for service providers by increasing compliance costs and changing the regulatory landscape [32].

### 1.2. A Call for Responsible, Smarter, and Sustainable Service Economies

Nevertheless, the technological progress achieved so far, while commendable, is lacking the level of responsibleness, smartness, and sustainability needed to tackle the challenges of our time and into the future [36,37]. Thus, there is an urgent need for innovative approaches for smarter and more sustainable services and service economies that can support the formation of sustainable cities and societies. We elaborate on this further below.

The service sector and service economies have become increasingly important to modern societies. However, they face a range of challenges, and their impacts on society and the economy are significant and unpredictable [38,39]. The situation requires careful management and adaptation to remain competitive and resilient in the face of changing economic, social, and environmental conditions [40,41]. It is essential for all stakeholders to collaborate and develop sustainable and responsible service practices [13,42]. This can be achieved by holistically and locally understanding the diverse needs and expectations of customers and the complex interrelationships among service providers, customers, and the environment [31,43]. For instance, in the context of the globalization challenges mentioned earlier, by studying service economies locally, policymakers and businesses can better understand and address these challenges, considering the specific needs and cultural contexts of local communities and working towards sustainable and responsible practices.

Therefore, a dynamic and interactive comprehension of the service sector and its related cultural, national, regional, and global issues is necessary to foster improvements and innovation on the way to smart and sustainable services. Furthermore, a holistic approach to service can help address issues such as income inequality, climate change, and demographic shifts.

Additional information regarding the gap in research is available in Section 2, while Section 6 provides a discourse on the originality, theoretical contributions, and practical applications of this study.

### 1.3. Scope of the Study

The main aim of this research has been to advance a methodology that allows for a comprehensive understanding of the service sector using artificial intelligence and that will drive future research in this field. Our ultimate aim has been to create a theory and approach for smarter and more sustainable services and service economies that can support sustainable future societies through the development of autonomous systems using innovative technologies and solutions (we elaborate on this in Section 6).

This research employs a data-centric methodology to comprehensively model the service sector. To achieve this, we utilize a combination of the academic literature and public opinion gathered from X (formerly known as Twitter). By leveraging advanced technologies such as deep learning and big data, we extract key parameters from both sources to gain a unique understanding of the service sector from two distinct perspectives. Although these perspectives can influence each other, they exhibit significant differences. To enhance our holistic understanding of service economies, additional perspectives can be gained from various data sources, including industrial news (offering an industrial perspective) and government documents (providing a governmental viewpoint).

We have created a software tool that utilizes our data-driven approach and a range of machine learning (ML) and other technologies. To uncover the key elements of the academic view of services, we constructed a dataset using the Scopus database. This dataset consists of 175,918 article data, published in English during the January 2018–June 2022 period. After analyzing the academic dataset, 29 distinct parameters related to the service sector were identified, which we grouped into 6 main categories (called macro-parameters): education & learning, healthcare, transportation & mobility, smart society & infrastructure, digital transformation, and service lifecycle management. From 9 October 2022 to 13 November 2022, we procured a dataset from Twitter to illustrate how the general population views the service industry within Saudi Arabia. The objective is to

concentrate on concerns specific to the regional service industry and contrast them with global perspectives, and therefore, we limited the dataset to the region of Saudi Arabia. The dataset includes 112,862 tweets, and 11 parameters were revealed through our tool, which were structured into 2 wider sets of parameters (macro-parameters): private sector services and government services. Details about the software tool can be found in Section 3.

The generated multi-dimensional taxonomy shown in Figure 1 provides a comprehensive overview of the service sector, incorporating multiple perspectives from academia and the general public. It offers a global outlook as well as a localized perspective specific to Saudi Arabia, highlighting the sector's diversity. The academic and global perspective helps to identify a wide range of services in the service sector, including education, healthcare, transportation, smart society and infrastructure, digital transformation, and service lifecycle management. It provides insights into the use of technology to improve service delivery and develop new business models, and it offers a comprehensive understanding of the current trends and developments driving the evolution of the service sector. Conversely, the local and public view provides insights into the types of services offered by the private sector and government in the Kingdom of Saudi Arabia (KSA). The private sector services include food delivery apps and services, logistics services, and others. The government's services include workforce development services, digital transformation services, and others. These services help to provide a comprehensive understanding of the types of services available to the public and how they are being delivered by both private and public entities. Understanding the availability, quality, and accessibility of these services can help policymakers identify gaps and develop strategies to improve the overall service delivery in the region.

The article exhibits a frame of knowledge, taxonomies, and a framework of the service sector generated with the help of the software tool, in addition to a detailed literature review based on over 300 research articles.

Figure 2 provides a framework for the autonomous design and operations of the service sector. It was developed using the parameters and information discovered in this paper. It is a systematic approach to automating services using data analytics, AI, machine and deep learning, and robotic systems to improve service quality and efficiency, reduce costs, and enhance customer satisfaction while ensuring data security and compliance with ethical and legal considerations. The framework encompasses a circular pattern that illustrates the iterative and continuous nature of the autonomous system design and operations process. It involves identifying services to be automated, defining scope and objectives, developing architectures, choosing the right technology, ensuring interoperability, implementing cybersecurity measures, establishing monitoring and feedback mechanisms, addressing ethical and legal concerns, implementing service lifecycle management, and developing a roadmap for implementation. Through the discovery of relevant parameters, the framework provides a comprehensive approach to the design and effective operation of autonomous systems for the service sector. The framework is further elucidated as we explore the specific parameters discovered throughout this paper.

The rest of the article is structured as follows. In Section 2, the relevant literature is explored, and gaps in research are identified. Section 3 elaborates on the proposed tool's research methodology and design. Section 4 investigates the academic and international parameters identified from academic data. Section 5 evaluates the public parameters identified through Twitter data. A discussion is presented in Section 6. Section 7 concludes the paper by summarizing key findings and indicating potential research opportunities.

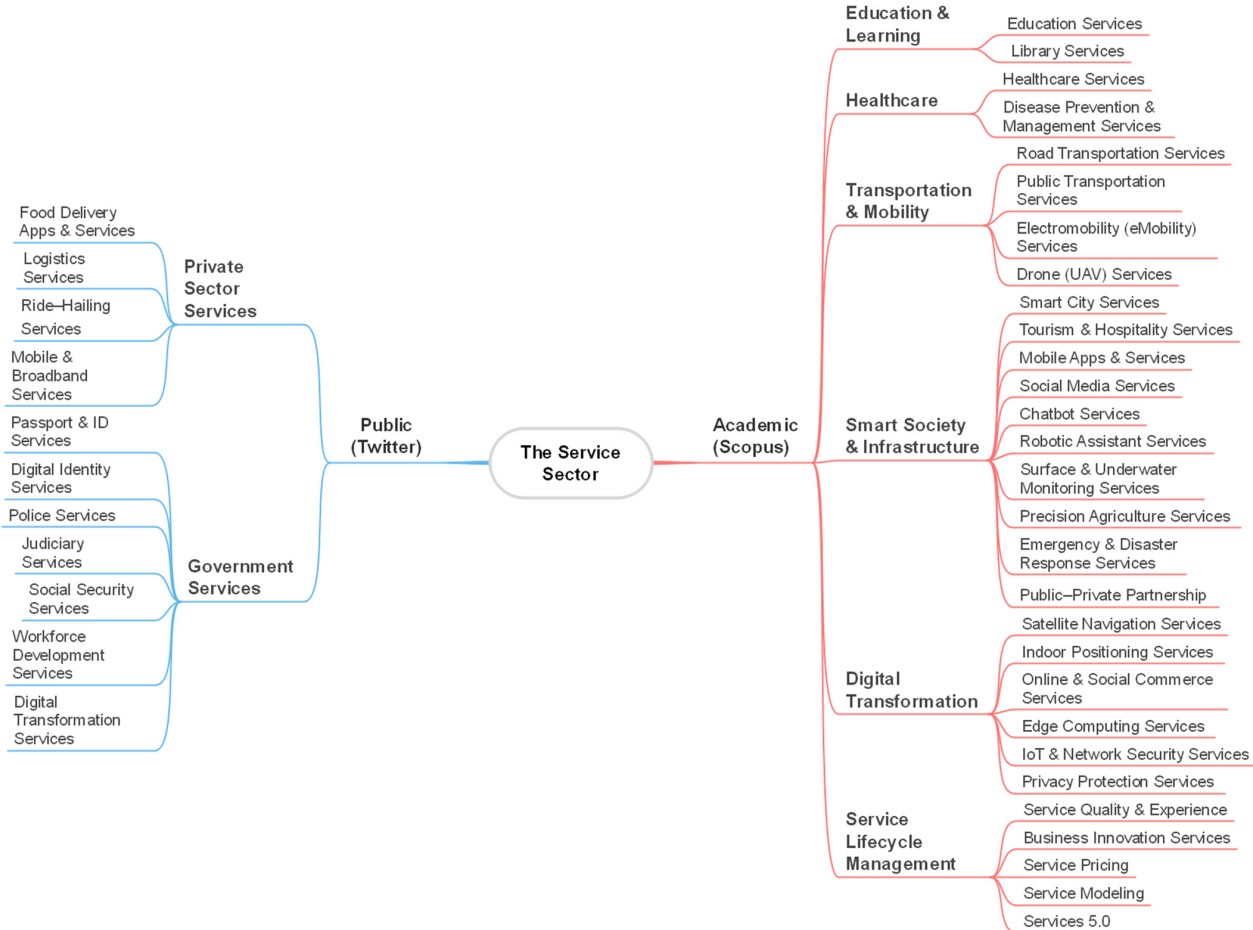

**Figure 1.** The service sector taxonomy providing a multi-perspective view dataset.

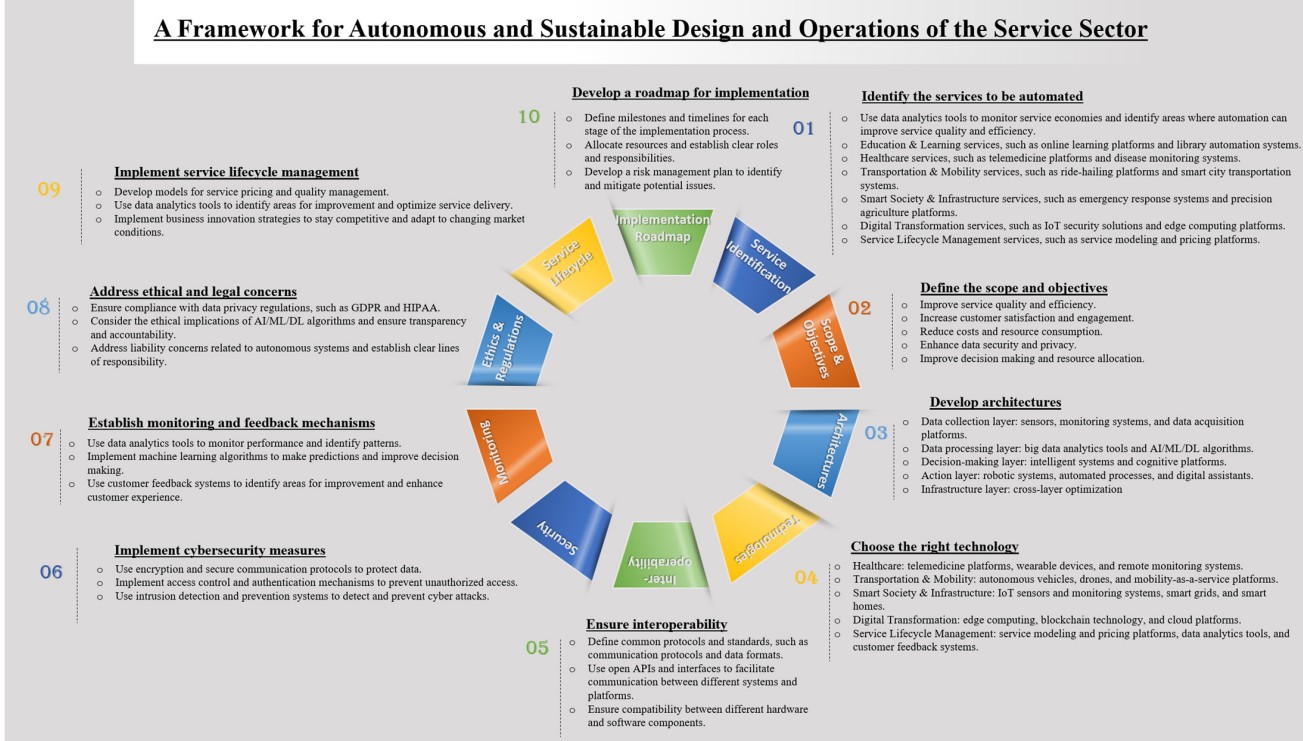

**Figure 2.** A framework for autonomous and sustainable design and operations of the service sector.

## 2. Related Works

This section reviews works related to our study. Our approach leverages machine learning to holistically map research in the service sector using a dataset of 175,918 papers from the Scopus database. Nevertheless, no research was found (in such a large dataset of nearly 200 K papers) that would directly relate to our study, which aims to discover holistic and multi-perspective parameters of the service sector using machine learning. Thus, we contextualize our work by discussing related research from two fields. The first field pertains to analyzing the scientific literature on services using scientometrics, as our work is based on analyzing the scientific literature. The second field relates to research on the use of machine learning for social media analysis in the service sector, which is relevant due to Twitter being a source of data in this work. The gaps in research are discussed next.

### 2.1. Service Sector Analytics Using Data from the Academic Literature

Bibliometric and scientometric analyses are widely employed for evaluating research in diverse areas, such as construction [44], finance [45], labor markets [35], explainable artificial intelligence [46], transportation [47,48], and smart families [49]. Notably, these analytical tools are also used in the service sector to study and comprehend the current literature on various topics related to services. For example, Ali et al. [50] identified trends in healthcare service quality, Arcese et al. [51] examined the family business model's role in the tourism sector, Dubina et al. [52] explored customer loyalty to banking services, and Vaz et al. [53] investigated the impact of sustainability and innovation on the automotive industry. Chang et al. [54] developed a paper categorization system in the field of tourism; Alsahafi et al. [31] investigated the tourism sector using Scopus data; Yas et al. [55] explored customer loyalty, satisfaction, and service quality in the marketing sector; and Ozyurt and Ayaz [56] studied the effect of publications from a journal on education.

Some work on data-driven design of services and product service systems is discussed below. Sassanelli et al. [57] highlight the rise in product-service systems and smart products, emphasizing the challenge for manufacturers to acquire knowledge and introducing a solution—the lean design rules tool (LDRT)—to enhance PSS design knowledge sharing. Pezzotta et al. [58] introduce the Product Service System Lean Design Methodology (PSSLDM), providing a structured approach for manufacturers adopting servitization and emphasizing integrated product and service design throughout the PSS lifecycle, with reported benefits by implementing companies. Additionally, Sassanelli et al. [59] delve into the transition from product to solution-based sales in digitalized manufacturing, focusing on integrating intangible services and introducing "lean PSS" through a literature review to define its features and research prospects.

While these studies have provided valuable insights into specific areas of research, such as healthcare, tourism, marketing, and education, there is a lack of holistic focus on the service sector. Each study has focused on a particular aspect of the service sector rather than investigating the sector as a whole. This suggests that there is a need for more comprehensive studies that consider the various dimensions of the service sector. By conducting such studies, researchers can gain a better understanding of the service sector as a whole, which can contribute to making informed policy decisions and business strategies.

### 2.2. Service Sector Analytics Using Social Media Data

Information obtained from online platforms, particularly Twitter, has been extensively utilized through textual analysis to explore diverse issues and subjects across numerous fields, including transportation [60], city logistics [61], COVID-19-related studies [62], education [63], smart homes [49], and healthcare [64]. Similarly, a considerable number of studies have applied textual analysis to Twitter data in the service industry. For instance, researchers have explored various service-related themes using social media data. Alahmari et al. [12] proposed co-creating cancer-related health services on Twitter. Zhan et al. [65] developed a framework to identify the most discussed topics concerning pharmacy organizations in the UK. Lee et al. [66] proposed a method for evaluating service quality in public

health. Ibrahim and Wang [67] analyzed Twitter to identify customer opinions and concerns about retail brands. Tian et al. [68] attempted to predict service quality in the airline industry. Numerous studies have explored the use of social media data to enhance service quality across various industries, including the transport sector [69], retail sector [70], food supply chains [71], food delivery applications [72], government policy development [73], and customer relationship management [74]. These studies demonstrate the potential of social media data and analytics to improve services across various industries. Despite several studies that have attempted to investigate different aspects of the service industry using social media data, none have done so holistically and nationally. Moreover, there have been only a limited number of studies that have utilized social media to investigate various service sectors in the Arabic language. These studies have predominantly focused on specific service sectors.

*2.3. Research Gap*

The analysis of the existing research literature shows that the service sector has received significant attention from researchers, with a particular focus on examining specific service sectors, service quality, and other elements that contribute to the overall service experience. While the current literature offers a wealth of information and insights into various aspects of the service sector, there is a need for a more comprehensive understanding of the landscape that considers the potential benefits of emerging technologies that can facilitate holistic optimization across different systems and sectors.

We seek to fill this void by offering a comprehensive analysis and visualization of the service industry landscape from a range of international and national perspectives. Our approach integrates advanced techniques in machine learning to facilitate the discovery and interpretation of knowledge and data. Unlike earlier works, which have focused on specific aspects of the service sector, our research takes a more holistic approach that examines the service sector as a whole and seeks to identify design and operation parameters that can improve overall efficiency and effectiveness.

A key contribution of our work is the development of a deep learning tool that can discover the structure and taxonomy of information related to the service sector, as well as identify opportunities for optimization across different systems and sectors. This tool is based on a rigorous methodology that draws on a range of data sources and analytic techniques, including bibliometric analysis and social media data analytics. The datasets used in this work have been gathered/developed from scratch, and no such datasets have been reported in the literature.

Moreover, aside from furnishing an all-encompassing perspective of the service sector landscape, our work also offers a review of the current literature on the service sector, highlighting key findings and insights that can inform policymaking and other decision-making processes. Using the discovered information, we also develop a comprehensive framework for the autonomous design and operations of service sectors using data analytics, AI, machine and deep learning, and robotic systems to improve efficiency, reduce costs, enhance customer satisfaction, and ensure data security and compliance with ethical and legal considerations.

In brief, our research contributes to a deeper understanding of the service sector and offers new insights and opportunities for optimizing the sector using cutting-edge technologies and analytical methods.

## 3. Methodology and Study Design

This section delves into the intricate details of our methodology and design. Figure 3 is the software architecture of our system, with its four components depicted in four vertical blocks. The architecture is expounded on in this section. From Sections 3.1–3.6, we provide an overview of the system and then discuss the various phases involved in the research and the tool.

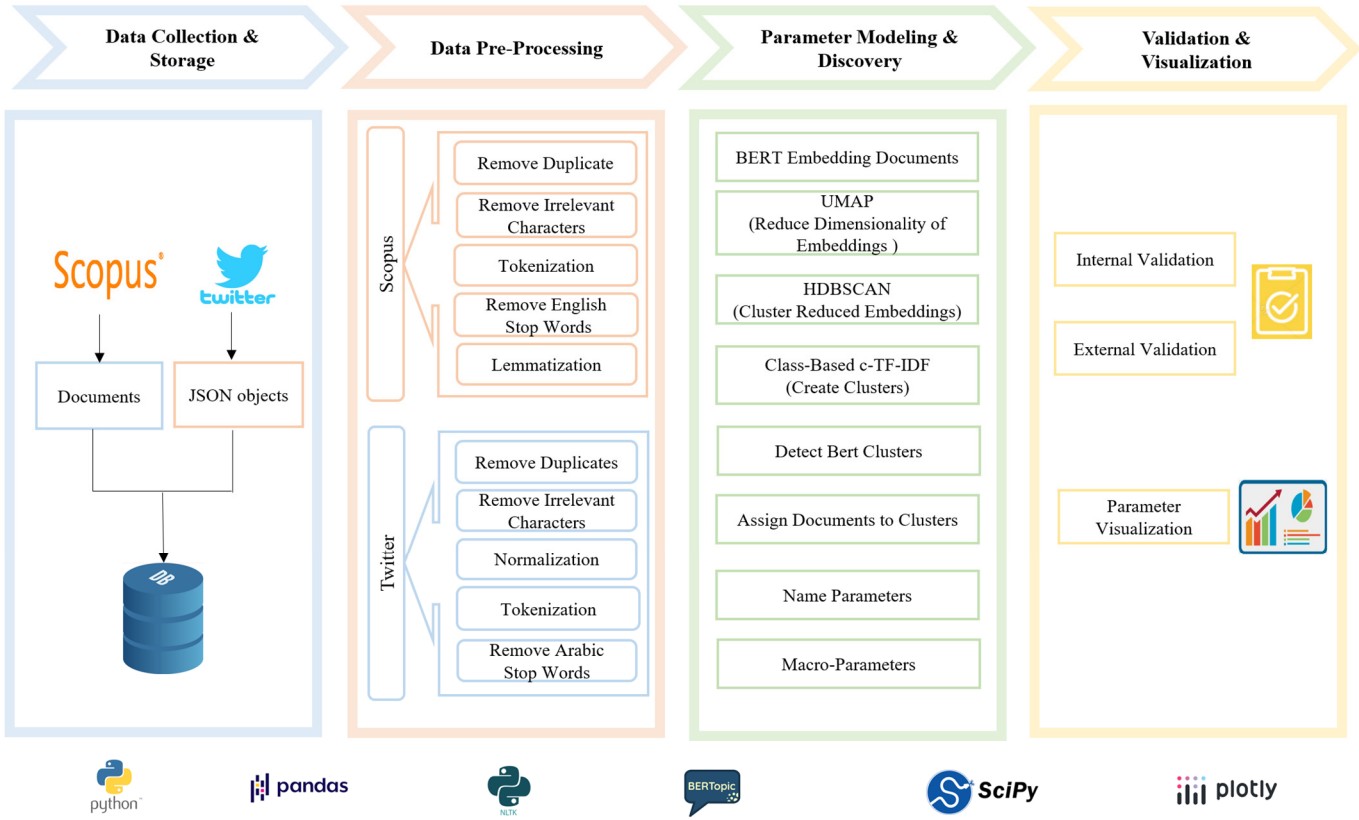

**Figure 3.** System architecture.

### 3.1. System Overview

Our dataset was created using a predefined search query, and the information was then saved in the CSV format. The data are then processed before the main processing with Pandas to implement the data pre-processing functions, such as removing duplicates from the dataset, removing characters that are irrelevant, accomplishing tokenization, and eliminating stop words from the dataset, as well as implementing lemmatization and then producing cleaned tokens. After that, we used the BERTopic model in our experiment with word embedding representations. The pre-trained BERT word embedding model has been utilized to identify the contextual relationships among words [75]. Then, we invoked UMAP to reduce embedding dimensions [76] and HDBSCAN to cluster semantically similar data together [77]. Using domain knowledge, hierarchical clustering, similarity matrix, and different quantitative analysis techniques, the clusters are then investigated, retagged as parameters, and further combined around macro-parameters. Moreover, the parameters were visualized using different numerical analysis techniques. Subsequently, validation of the identified parameters based on the scientific literature, tweets, and other sources is performed.

### 3.2. The Dataset

For this dataset, we gathered data from Twitter for the public perspective and data from Scopus for the academic viewpoint. Since Scopus is a database that indexes scholarly research, we believe that it offers an academic perspective on the service sector. Although public perspectives and circumstances could be described in academic publications, as they are viewed and voiced by academics, these perspectives could still be regarded as academic. In addition, we used Twitter to learn what the general population thinks about services. Twitter may contain posts from different stakeholders; therefore, it might be used to capture the perspectives of other stakeholders. Nonetheless, the tweets from diverse stakeholders are primarily intended to interact with people.

### 3.2.1. Scopus Data

We captured the research articles related to the service sector in the English language from Scopus. In this research, we gathered 175,918 research articles for the period 2018–2022 (collected until June 2022, the time of data collection) using the term "service" and the subject area "Computer Science". The final dataset, after cleaning and removing duplicates and irrelevant papers, comprised 42,389 documents. The document types of the collected data were not restricted to any type and included journal articles, etc.

### 3.2.2. Twitter Data

The Twitter REST API was used to acquire the Arabic tweets. Tweets were collected by using different key terms related to services. For example, we used the key terms transport ("خدمات النقل") (Najiz service), "خدمة ناجز" (government service), "خدمة حكوميه" services), "تطبيقات التوصيل" (delivery apps), "الاقتصاد الرقمي" (digital economy), and others. We exclusively collected tweets sent within Saudi Arabia using geolocation filtering to find important services pertaining to Saudi society. The total number of tweets collected from Twitter was 112,862 tweets during the period from 9 October to 13 November 2022. The Twitter data were in the form of tweet objects in JSON with a series of attributes such as "created_a", "text", and others. We converted the collected JSON file into CSV format. Also, the dataset was cleaned by removing duplicates, resulting in a total of 84,271 tweets.

### 3.3. Data Pre-Processing

### 3.3.1. Scopus Data Pre-Processing

Data pre-processing involves various steps to data manipulation or deletion before usage in order to guarantee or improve performance. Data pre-processing includes the following steps: remove duplicate articles, remove irrelevant symbols, tokenize, remove English stop words, and lemmatize. First, using the Python library "Pandas", we read the CSV file into a data frame. To avoid the frequency of the same articles, we removed duplicate articles. Then, we removed all irrelevant digits, characters, and extra spaces. Fourthly, we implemented tokenization on the dataset using tokenize() to split a string into a list of words. The step that followed involved removing the English stop words using a predefined list of stop words, a natural language toolkit (NLTK). Finally, we implemented the lemmatization process using the WordNet Lemmatizer module. Consequently, we received and stored the cleaned articles.

### 3.3.2. Twitter Data Pre-Processing

The text posted on social media is unstructured data that needs cleaning. The main data pre-processing steps are as follows: remove unessential characters, normalize, tokenize, and remove stop words. We loaded the tweets into Panda Data Frames. To avoid the frequency of the same tweets, we removed duplicate tweets. We removed extra spaces, lines, emails, repeated characters, single quotes, the English alphabet, and all emojis from the tweet. Additionally, we removed the punctuation by making a list of all punctuation, such as semi-colons (;), periods (.), colons (:), question marks (?), and others. Also, we removed diacritic marks such as Fatha (َ), Kasra (ِ), Damma (ُ), Tanwin Fath (ً), Tanwin Kasr (ٍ), Tanwin Damm (ٌ), Sukun (ْ), and Tashdid (ّ). This step has made information more valuable and decreased the size of the feature set.

Then, we implemented tokenization on the dataset using tokenize() to split each tweet into a list of tokens. The step that followed involved normalizing the letters that have multiple shapes into a consistent form, such as the Arabic letter Alif (أ). We normalized three letters (آ ا إ) to (ا). Also, the letter Yaa (ي) was normalized to (ى), and Taa Marboutah (ة ه) was normalized to Taa (ة). Then, we removed stop words that were not important in extracting the information. We used NLTK to remove the stop words by adding a new list of stop words used in dialectical Arabic, including "انتو" ,"وين" ,"علشان" ,"بينما", and

"بيني". Before implementing Bert topic modeling, we used clustering to obtain the tweets that are related to services in Saudi Arabia alone. We used a list of service keywords to filter the tweets. For example, we used the key terms: "اقتصاد" (economy), "التوصيل" (delivery), "الشحن" (shipping), "النقل" (transport), "اللوجستية" (logistics), "القضاء" (judiciary), "الاتصالات" (telecommunications), and others.

### 3.4. Parameter Modeling

We employed BERTopic modeling to classify the collected dataset into topic clusters. BERT is used to understand language, and it is a pre-trained version of deep bidirectional transformers. We created a word embedding model using a transformer-based technique called BERT [75]. Contextualized word representations are extracted for all tokens using embedding models and subsequently fed into BERTopic. In our experiment, we used pre-trained language models of sentence transformers, called "distilbert-base-nli-mean-tokens", with the Scopus dataset, used for clustering and semantic search tasks. For the Twitter dataset, we used pre-trained language models of sentence transformers, called "aubmindlab/bert-base-arabertv2"; AraBERT is an Arabic pre-trained language model. After the embedding step, we performed a dimensional reduction algorithm, UMAP to reduce the dimension of the embedding. After that, we used HDBSCAN to cluster semantically similar clusters together. If we consider all documents in the cluster as a single document and implement TF-IDF, this has enabled us to obtain a significant score for each word in the cluster, which is named the c-TF-IDF score. When the words in a cluster carry greater importance, the parameter becomes a more accurate reflection of the group of words. So, we can find a set of keywords that describe every parameter.

Estimating the number of parameters to be retrieved from the documents prior to model training is a challenging task. Thus, to achieve optimal outcomes, BERTopic was trained on the documents, and a series of trials were executed. The most favorable model was then preserved. The domain knowledge of the authors, the information gained from the modeling, and their quantitative analysis were used to convert a set of clusters to a set of parameters and consolidate them into macro-parameters.

### 3.5. Quantitative Analysis and Visualization

BERTopic provides different visualization possibilities that help to obtain more insight into each topic. These quantitative analysis methods with domain knowledge contribute to discovering the parameters and macro-parameters. The main methods of quantitative analysis are discussed as follows. Inter-topic Distance Map: It can be described as a two-dimensional representation of topics. In BERTopic, an inter-topic distance map can be visualized by using the visualize_topics() function to represent the topics that were generated as circles with the top 10 most frequent terms for each topic and the distance between topics. Keyword Score: A BERT model is given a list of keywords that describe the parameters. Each keyword has to explain the context of the parameter and represent the value of c-TF-IDF (see Section 3.4). Term Score: Term score is a method for the visualization of the ranks of all terms (keywords) for all topics. Each topic is represented with a set of terms in terms of how representative they are of the topic in terms of the c-TF-IDF scores. A higher score indicated a higher representation. A term score graph depicts the scores of c-TF-IDF for every topic with each word's term rank. Hierarchical Clustering: Hierarchical clustering helps to visualize the hierarchical structure of the topics via systematically pairing clusters and visualizing how they relate to one another. We used Ward's linkage function to implement hierarchical clustering using a cosine distance matrix between topics. Similarity Matrix: It is visualized using the matrix of cosine similarity among topic embeddings. We utilized the Plotly library to visualize the heatmap using a similarity matrix. To demonstrate the relationship between the topics, the similarity matrix was generated by calculating the cosine similarity between the topics. We have used the continuous color scale "BnGu",

green to blue, from Plotly, where clusters with the highest similarity are represented with dark blue, while those with the least similarity are depicted in light green.

### 3.6. Evaluation and Validation

To validate our results for the proposed system, we used two different approaches: internal and external validation. The internal one is implemented by determining whether the documents involving academic articles or tweets are associated with a specific parameter. We define and expound upon the parameters by drawing insights from related academic articles as well as pertinent tweets. The external validation is achieved by performing comparisons for taxonomies and quantitative data between the two datasets. We also utilized different online sources to validate the information about the parameters. Additionally, we performed both types of validation using visualization. These visualization methods allow for elaboration on the datasets and the discovered services, such as taxonomies, temporal progression plots, similarity matrices, and others. A number of Python libraries have been used with our tool to perform the visualizations, such as Plotly, Matplotlib, and others.

## 4. Global Analysis Results: Service Sector Parameters from Scopus

This section describes the parameter space of the parameters that were extracted using our tool from the Scopus data. The parameter space is discussed in Section 4.1, followed by a quantitative analysis in Section 4.2. Sections 4.3–4.8 go into more depth about each specific macro-parameter. For each macro-parameter, we describe it using its clustered articles and sources external to the dataset.

### 4.1. Overview

Using our tool and the Scopus dataset, we identified a total of 30 clusters, numbered 0 to 29. One of the 30 clusters was unrelated to the topic of the work, cluster number 12, so we excluded it from the analysis. On the basis of domain expertise and quantitative techniques, we identified and labeled 29 parameters for the service sector, categorized into 6 macro-parameters. The macro parameters are listed in Column 1 of Table 1, namely education & learning, healthcare, transportation & mobility, smart society & infrastructure, digital transformation, and service lifecycle management. The names of each parameter and their associated cluster numbers are presented in Columns 2 and 3. In Column 4, we listed the percentages of the articles for each parameter. Column 5 includes the top ten parameter-affiliated terms. The taxonomy presented in Figure 4 illustrates the map of the parameter space in the service sector, as detected with our tool. The classification was derived from the information presented in Table 1. The topmost layers of the taxonomy represent the macro-parameters, followed by the parameters in the middle layers, and the dimensions associated with each parameter in the lowermost layers.

**Table 1.** The service sector parameter space providing an academic view.

| Macro-Parameters | Parameters | Cluster No. | % | Keywords |
|---|---|---|---|---|
| Education & Learning | Education Services | 4 | 2.14 | Student, Learning, Education, Teaching, University, Course, Educational, School, Online, Research |
| | Library Services | 23 | 0.55 | Library, Research, Service, Data, University, Academic, Digital, Science, Community, Project |
| Healthcare | Healthcare Services | 0 | 8.14 | Patient, Health, Healthcare, Medical, Care, Data, Hospital, System, Disease, COVID |
| | Disease Prevention & Management Services | 22 | 0.57 | Web, Available, Tool, Data, Disease, Analysis, Database, Prediction, Structure, Site |

**Table 1.** *Cont.*

| Macro-Parameters | Parameters | Cluster No. | % | Keywords |
|---|---|---|---|---|
| Transportation & Mobility | Road Transportation Services | 10 | 1.42 | Vehicle, Taxi, Car, Road, Driver, Traffic, Driving, Safety, Transportation, Service, System |
| | Public Transportation Services | 14 | 0.88 | Bus, Passenger, Train, Railway, Rail, Transit, Stop, Travel, Track, Line, Station |
| | Electromobility (eMobility) Services | 8 | 1.85 | Energy, Charging, Electricity, Renewable, System, Electric, Wind, Demand, Control, Market |
| | Drone/UAV Services | 24 | 0.55 | Drone, Aircraft, Airport, Flight, Air, Ground, Aviation, Passenger, Service, System |
| Smart Society & Infrastructure | Smart City Services | 11 | 1.22 | City, Smart, Urban, Data, Citizen, Technology, Service, Development, Public, Sustainable |
| | Tourism & Hospitality Services | 28 | 0.46 | Tourism, Tourist, Hotel, Industry, Service, Information, Customer, Travel, Review, Development |
| | Mobile Apps & Services | 7 | 1.87 | Mobile, User, Apps, Smartphone, Smartphones, Device, App, Android, Privacy, Application |
| | Social Media Services | 17 | 0.78 | Social, Twitter, Tweet, Medium, User, Facebook, Sentiment, Opinion, Information, News |
| | Chatbot Services | 21 | 0.58 | Chatbots, Chatbot, Customer, Service, User, Research, Factor, Conversation, System, Data |
| | Robotic Assistant Services | 6 | 1.88 | Robot, Human, Robotic, Object, Task, Robotics, Environment, System, Service, Autonomous |
| | Surface & Underwater Monitoring Services | 27 | 0.48 | Water, Marine, Sea, System, Sensor, Data, Monitoring, Acoustic, River, Ocean |
| | Precision Agriculture Services (PAS) | 19 | 0.66 | Agricultural, Forest, Agriculture, Crop, Soil, Data, Farm, Land, Production, Rural |
| | Emergency & Disaster Response Services | 15 | 0.88 | Disaster, Flood, Weather, Emergency, Fire, Earthquake, System, Event, Response, Data |
| | Public–Private Partnership (PPP) | 13 | 0.94 | Service, Process, System, Government, Information, Design, Approach, Data, Research, Management |
| Digital Transformation | Satellite Navigation Services | 16 | 0.84 | Satellite, Orbit, Navigation, GNSS, Bd, Positioning, Signal, System, Constellation, Earth |
| | Indoor Positioning Services (IPS) | 20 | 0.65 | Indoor, Positioning, Localization, Location, Fingerprint, Accuracy, Signal, Environment, System, Navigation |
| | Online & Social Commerce Services | 18 | 0.68 | Customer, Online, Review, Commerce, Consumer, Product, Social, Service, Brand, Purchase |
| | Edge Computing Services | 9 | 1.56 | Network, Resource, Service, Reduce, Problem, Cost, Task, Energy, Computing, User |
| | IoT & Network Security Services | 2 | 3.51 | Attack, Detection, Network, Denial, Security, Do, Traffic, Malicious, Detect, Threat |
| | Privacy Protection Services | 5 | 2.07 | Privacy, Data, Security, User, Encryption, Cloud, Authentication, Key, Secure, Encrypted |

**Table 1.** *Cont.*

| Macro-Parameters | Parameters | Cluster No. | % | Keywords |
|---|---|---|---|---|
| Service Lifecycle Management | Service Modeling | 25 | 0.54 | Queue, Customer, Arrival, Distribution, Queueing, System, Service, Probability, Time, Packet |
| | Service Pricing | 29 | 0.44 | Price, Profit, Customer, Service, Pricing, Provider, Market, Optimal, Consumer, Cost |
| | Service Quality & Experience | 1 | 5.07 | Wireless, Network, Communication, Mobile, Energy, User, Service, Performance, Transmission, Access |
| | Business Innovation Services | 26 | 0.52 | Business, Digital, Product, Company, New, Process, Industry, Service, Manufacturing, Innovation |
| | Services 5.0 | 3 | 2.30 | Cloud, Home, IoT, Data, Service, Smart, Application, Computing, Device, Internet |

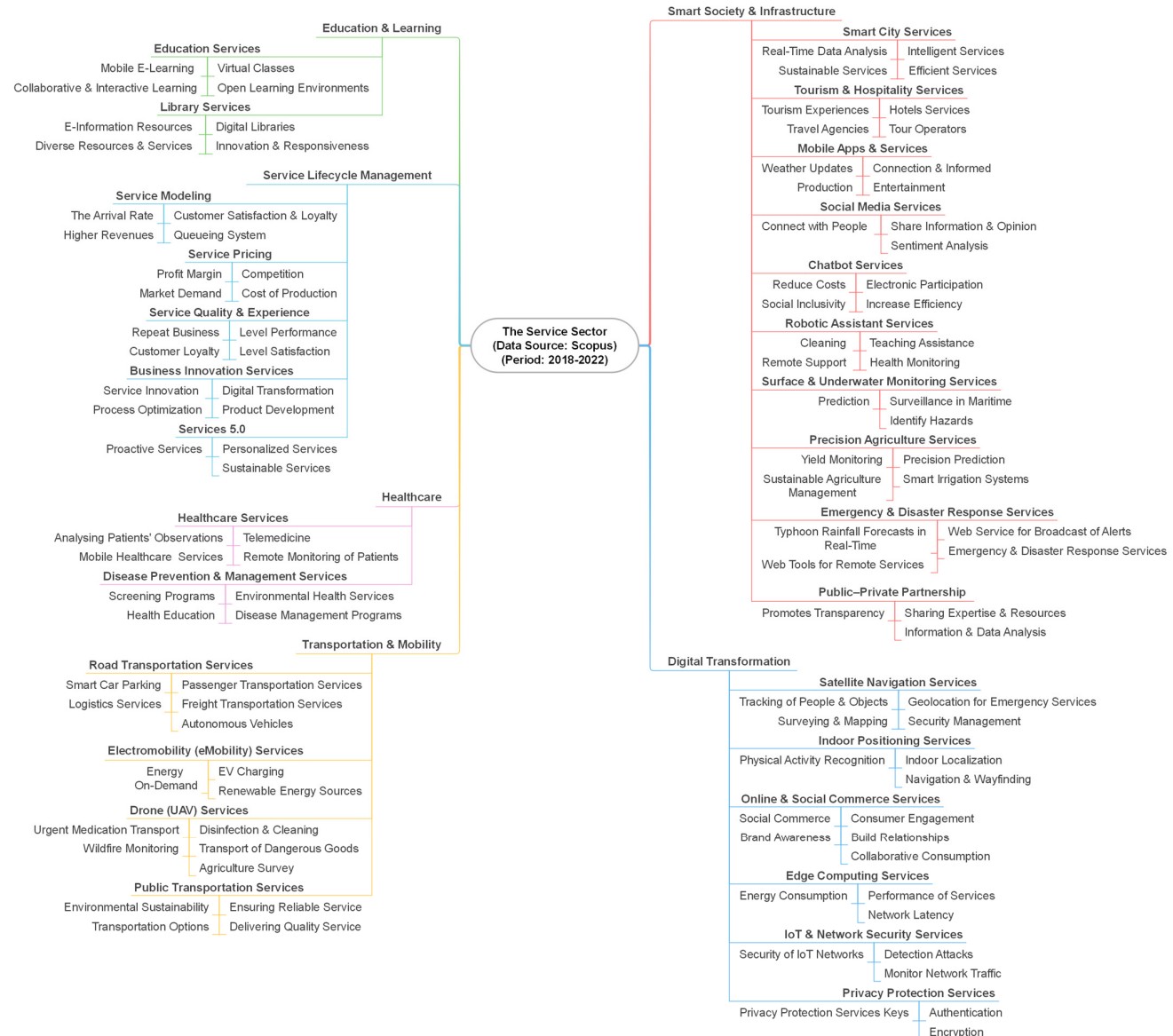

**Figure 4.** The service sector taxonomy providing an academic view.

*4.2. Quantitative Analysis (Scopus Dataset)*

A quantitative analysis of the results has been carried out using our software tool and the Scopus dataset. The analysis includes a term score decline graph, an inter-topic distance map, a similarity matrix, keyword scores, and a hierarchical clustering map. To keep this brief, we have omitted these graphs, but they are available upon request.

*4.3. Education & Learning*

This macro-parameter encompasses all services related to education & learning, from traditional classroom education to online learning and educational support services [78,79]. The goal of these services is to provide learners with the skills and knowledge they need to succeed in their personal and professional lives. Education services can include everything from K-12 education to post-secondary education, vocational training, and adult education. Library services are also included in this macro-parameter, as libraries serve as important resources for learners of all ages by providing access to information and knowledge.

4.3.1. Education Services

Education services are professional support provided by educators or educational institutions to facilitate personal and professional growth through teaching, curriculum development, assessment, technology implementation, and advisory services [80]. The parameter encompasses various dimensions of education services, including the importance of open learning environments for reducing geographical and time limitations [78], the use of modern technology in delivering virtual classes [81], mobile e-learning [82], the impact of online social networks on students' development [83], educational sustainability in general, and, in particular, the use of social media platforms and modern technologies for educational sustainability during pandemics and disasters [84,85]. Other dimensions include improving learning outcomes [82,86], continuing out-of-class communication (OCC) [83], evaluating the quality of services in education systems [81], enhancing self-efficacy for online learning [87], overcoming the barriers of traditional education [86], and promoting collaborative and interactive learning [88]. For instance, regarding education sustainability, Baytiyeh [84] highlighted how social media platforms can contribute to educational sustainability and accessibility during times of disaster when educational institutions are unable to guarantee students' safety or provide secure learning conditions.

4.3.2. Library Services

Library services are a collection of resources, facilities, and support provided by a library to meet the information and educational needs of its users [89,90]. Library services support research and learning, promote literacy, and enrich the cultural and intellectual life of the community [91,92]. Library services are diverse and aim to provide quality standards and efficient delivery of information to support learning and research goals [93,94]. This parameter is characterized by various dimensions of library services. These dimensions include access to diverse resources and services, both physical and digital, effectiveness in meeting user needs, innovation and responsiveness to changing user needs and new technologies, collaboration with other institutions and stakeholders, outreach to the wider community, quality standards in terms of resources, expertise of staff, and user experience, and providing electronic information resources through digital libraries to support e-learning [95–98]. For example, see [99] (use of mobile augmented reality (MAR) technology to access library services) and [100] (the role of libraries in supporting e-learning).

*4.4. Healthcare*

This macro-parameter encompasses all services related to healthcare [101,102], including preventative care, treatment of illnesses and injuries, and management of chronic conditions. The goal of healthcare services is to maintain and improve the health of individuals and communities [103]. Examples of healthcare services include hospitals, clinics, medical practices, and public health organizations. Disease prevention and management

services are also included in this macro-parameter, as they play a crucial role in maintaining the health of individuals and communities.

### 4.4.1. Healthcare Services

Healthcare services encompass a wide range of medical and therapeutic interventions, provided by healthcare professionals and facilities, aimed at diagnosing, treating, and preventing illness, injury, and disease through primary care, specialized care, diagnostic services, and therapeutic interventions [12]. Healthcare organizations and service providers endeavor to improve the quality of services provided to patients [104] and analyze and track patient data to improve healthcare services [101,105].

This parameter helped to cover multiple dimensions of healthcare services, including telemedicine for remote healthcare delivery [106,107] and ICTs for disease detection and prediction [64,108]. Additionally, it highlights the importance of providing healthcare services in rural and remote areas [106] and engaging citizens in co-creating value for services [12].

### 4.4.2. Disease Prevention & Management Services

Disease prevention and management services include interventions to prevent or reduce the impact of diseases on individuals and communities [64]. These services comprise immunizations, health education, screening programs, disease management programs, and environmental health services [12]. These services are vital for promoting healthy communities and reducing the burden of illness by providing education, immunizations, screenings, and disease management programs. In the age of digital technology, the web has become a vital platform for accessing and sharing information. One area where the web has proved particularly useful is in disease analysis and prediction. Thanks to the availability of data and the development of powerful tools for data analysis, researchers and healthcare professionals can now use digital and web-based databases and sites to gather, store, and analyze vast amounts of health-related information [62]. These databases and sites are structured to enable efficient and accurate data retrieval and analysis, making it easier to identify patterns, trends, and risk factors associated with different diseases [109]. Using these tools, researchers and healthcare professionals can develop predictive models that can help anticipate and prevent disease outbreaks, ultimately leading to better health outcomes for individuals and communities [110,111].

This parameter covered multiple dimensions of these services, including IoT, smart farming systems, disease prediction using social media and networks, activity recognition, and smart homes. These technologies allow for remote monitoring, timely interventions, personalized prevention and management plans, and healthier living environments [112,113].

### *4.5. Transportation & Mobility*

This macro-parameter encompasses all services related to transportation & mobility [114], from traditional road transportation to innovative services such as electromobility and drone services. The goal of these services is to provide individuals and goods with safe, efficient, and reliable transportation options [115]. Examples of transportation services include public transportation services such as buses and trains, ride-sharing services such as Uber and Lyft, and eMobility services such as electric cars and bikes [116]. Drone services are also included in this macro-parameter, as they represent an emerging technology that has the potential to revolutionize the way goods are transported.

### 4.5.1. Road Transportation Services

Road transportation services involve the movement of people, goods, and services from one place to another using various vehicles, including cars, taxis, trucks, motorcycles, and other automobiles [117]. Drivers operate these vehicles on the road system, which can be affected by traffic congestion, weather conditions, and other factors that impact driving safety. Road transportation services play a crucial role in facilitating trade and

commerce, as well as providing access to essential goods and services. With the continued development of technology and innovation [118], these services are expected to continue to improve and enhance the quality of life for individuals and communities [119].

Our tool helped to cover multiple dimensions of these transportation services, such as passenger transportation and freight transportation, logistics, and warehousing. With the emergence of technology, road transportation services have evolved to include systems such as autonomous vehicles, connected vehicles, and the internet of vehicles (IOV) [120]. These systems aim to optimize traffic and improve safety by enabling information sharing and traffic optimization. For example, see the benefits of autonomous and connected vehicles in enhancing safety and reducing traffic congestion [118] and the internet of vehicles to improve road safety by exchanging information among vehicles to prevent accidents [121].

### 4.5.2. Public Transportation Services

This parameter covered various dimensions of public transportation services, including congestion reduction, resilience, performance, quality of service, and sustainability [122,123]. These dimensions encompass aspects such as providing effective transportation options, ensuring reliable service, delivering quality service, and maintaining environmental sustainability [48]. Public transportation services such as buses, trains, subways, and trams provide affordable and efficient transportation options for passengers in cities and regions. Buses have features such as air conditioning, wheelchair accessibility, and bike racks, while trains and subways offer faster and more efficient options with designated stations and tracks that provide higher passenger capacity. These services play a crucial role in reducing traffic congestion, promoting sustainable modes of travel, and providing access to transportation for a wide range of people [124,125]. For example, see an integrated system of wireless communication in urban rail transit with a high QoS to share data pertaining to train safety [126] and integrate QoS and energy savings parameters in urban rail transit systems [127].

### 4.5.3. Electromobility (eMobility) Services

This parameter helped cover a range of dimensions of electromobility (eMobility) services. The term eMobility refers to a range of products and services that facilitate the use of electric vehicles (EVs) as a mode of transportation [114,128]. These services play a crucial role in reducing greenhouse gas emissions and supporting the transition to a more sustainable energy system [129]. Electromobility services involve the development, installation, and maintenance of EV charging infrastructure and related services such as battery swapping, fleet management, and energy storage [130]. The EV charging infrastructure uses electricity, which can be sourced from renewable sources such as solar and wind, to meet the growing demand for EV charging. Smart charging solutions that optimize charging times based on energy demand and pricing help control electricity usage [131]. The integration of EV charging with renewable energy sources and the implementation of energy storage solutions for EV charging further help in the transition to a more sustainable transportation system. Various players in the market offer eMobility services, and the market for these services is growing as more people adopt EVs [48]. We discovered a significant amount of interesting research that relates to this parameter. For example, enhancing eMobility services and QoE for end users by processing content at the network's edges [132], EV charging services as a promising solution to address excessive fuel consumption and greenhouse gas emissions [133], eMobility services to aid in reducing environmental pollution [134], and promoting economic power systems that rely on renewable energy sources [135].

### 4.5.4. Drone/Unmanned Aerial Vehicle (UAV) Services

This parameter captured multiple dimensions of drone (or unmanned aerial vehicle (UAV)) services, including agriculture surveying, delivery services, wildfire monitoring, medical and healthcare services, transport of dangerous goods, disinfection and cleaning, and urgent medication transport. Drones and UAVs are terms that are used interchangeably,

though some may differentiate between them. Additionally, the terms RPAS and UAS are also sometimes used instead of drones or UAVs. While these terms all refer to unmanned aerial vehicles, they may also be used to specify certain features or applications of the aircraft [136]. Drones have revolutionized various industries by providing numerous benefits [137–141], such as improved safety, efficiency, and cost-effectiveness. However, there are concerns about their use near airports and controlled airspace, as well as in other areas such as residential neighborhoods, public spaces, and critical infrastructure. The potential risks include collisions with other aircraft, invasion of privacy, and disruption of public safety operations.

Examples of research related to this parameter include the following: the usefulness of drone fleets in large-scale agriculture and forestry surveying [142], the utilization of drones for remote wildfire monitoring [143], the benefits of deploying drones for medical and healthcare-related services, especially for patients in hard-to-reach areas [144], the significance of using drones for the safe transportation of dangerous goods in medical logistics [145], and the role of drones in the disinfection and cleaning of surfaces during COVID-19 [146].

### 4.6. Smart Society & Infrastructure

This macro-parameter encompasses a wide range of services related to the development of smart societies and infrastructure. These services utilize innovative technologies such as AI, machine learning, and IoT to create more efficient and effective ways of delivering services to individuals and communities [147]. Examples of services in this macro-parameter include emergency and disaster response services, surface and underwater monitoring services, tourism & hospitality services, public–private partnership services, and social media and chatbot services. These services are designed to improve the overall quality of life for individuals and communities by providing them with more effective and efficient services [148].

#### 4.6.1. Smart City Services

Smart city services refer to a set of advanced technological solutions aimed at making urban areas more intelligent, efficient, sustainable, and livable [149]. Smart city services enhance urban development by providing accessible, citizen-focused services that promote sustainability, economic competitiveness, and quality of life [150]. These services utilize data-driven development and real-time data analysis to provide accessible digital services that are responsive to the needs of citizens [151]. These services create a more responsive, efficient, and sustainable urban environment for citizens and are designed to be accessible to both inhabitants and city authorities, with the government, citizens, and businesses co-creating digital solutions that provide value to all stakeholders.

Smart waste management systems use sensors and real-time data analysis to improve waste recycling, garbage collection, and waste disposal [152]. Similarly, smart water management solutions leverage technology to manage water supply, reduce water waste, and promote sustainable development. Among the articles in this parameter, consider, for example, citizen participation in new ICT services and applications across various aspects of urban life to achieve sustainability, reliability, resilience, and smartness in cities [153], as well as the use of data-driven approaches to sustainable urban development to help engage companies and citizens in the process of monitoring, planning, and analyzing urban processes, optimizing urban services, and increasing the level of smartness and sustainability in cities [154].

#### 4.6.2. Tourism & Hospitality Services

Tourism & hospitality services refer to a range of businesses and organizations that provide services to people who are traveling or seeking leisure activities [155]. These services can include accommodation, food and beverage, transportation, entertainment, and other activities. Tourism & hospitality services are integral parts of the global travel and

tourism industry, which has experienced significant growth over the past decade (though there was a significant decline due to COVID-19 [31]).

Our tool helped to cover various dimensions of tourism & hospitality services that cater to the needs of tourists [156], including hotels, restaurants, transportation services, tour operators, and travel agencies. For example, utilizing vast amounts of data (i.e., big data) generated by visitors to offer personalized travel options and assist tourism stakeholders in making strategic decisions [157]; the contribution of tourism services to improve people's welfare through a creative process that drives the economy [158]; the role of smart tourism in significantly improving tourism management, economics, and services for the growth of the tourism sector [159]; and an in-depth recent review of the tourism sector [31].

### 4.6.3. Mobile Apps & Services

Mobile apps & services have become an essential part of modern life, enabling users to stay connected and informed, be more productive, and engage in a variety of activities on-the-go [160]. Mobile apps and services are software applications designed for mobile devices such as smartphones, tablets, and wearables [161]. These apps can be downloaded from app stores, which are online platforms operated by smartphone operating system providers or independent developers.

Mobile apps and services offer numerous benefits to users, including convenient access to a range of features such as social networking, entertainment [162], productivity, e-commerce [160], and education [163]. However, the use of mobile apps and services also raises privacy concerns due to the amount of personal information that they collect from users. Therefore, it is essential to implement privacy controls to ensure that users' personal information is protected. In addition, mobile apps and services may consume significant amounts of resources, such as battery life, data, and processing power. For instance, Ren et al. [164] explained that the EasyPrivacy system for Android smartphones provides an automatic context-aware resource usage control system when using numerous applications.

### 4.6.4. Social Media Services

This parameter covered several dimensions of social media services, which are online platforms that allow users to create and share information, news, and opinions, interact with others, and connect with people from all around the world [165]. Twitter is one such platform; it allows users to share their thoughts and opinions in the form of tweets [60].

Social media services offer diverse features, such as profiles, where users can provide information about themselves and their interests. Sentiment analysis is a technique that analyzes social media information to determine the opinions, attitudes, and sentiments expressed by users [166,167]. This can be useful in various fields, such as understanding customer satisfaction, analyzing online reviews, and assessing service quality. Businesses can use a sentiment analysis to make data-driven decisions based on consumer behavior and opinions expressed on social media.

The availability and accessibility of social media data and information have facilitated the spread of information. This has led to a shift in the way information is consumed and has influenced consumer behavior. The spread of social media information and news has also made it easier for businesses to monitor and analyze customer sentiment and opinions, which can inform their business decisions. For examples of research on this parameter, see [168,169].

### 4.6.5. Chatbot Services

A chatbot service is an innovative computer program designed to facilitate electronic participation and co-designing processes between businesses and their customers [170,171]. Chatbots are designed to simulate human-like conversations with users via messaging platforms or mobile devices [172–174]. These systems are powered by data and research, ensuring they can deliver high-quality customer service. Chatbots can be rule-based or AI-powered [175,176], with the latter being able to understand natural language and provide

more personalized and context-aware responses. Chatbots can help businesses reduce customer service costs and increase efficiency. They can also facilitate customer feedback and social commerce, enabling businesses to build strong customer relationships and promote sustainability. Additional examples of research on the topic include [177,178].

### 4.6.6. Robotic Assistant Services

This parameter covered multiple dimensions of robotic assistant services [179]. These robots are designed to perform specific functions using a combination of sensors, AI, and ML. Robotic assistant services offer numerous benefits, including increased efficiency, improved accuracy, and reduced labor costs. They provide assistance in a number of domains, including personal and professional tasks. For instance, personal assistant robots can help humans with scheduling, setting reminders, and organizing appointments. Eldercare robots can assist the elderly with medication reminders, physical therapy, and health monitoring [180]. Cleaning robots, on the other hand, can clean floors, windows, gutters, and other hard-to-reach areas, which can be challenging for humans [181]. Robots can also perform tasks in hazardous or challenging environments, reducing the risk of injury to humans. Teaching assistance robots can be used in educational settings to help students learn, interact, and communicate with others. Social interaction robots can help individuals with social skills training or assist those who suffer from mental health issues. Additional examples of studies that belong to this parameter include [182,183].

### 4.6.7. Surface & Underwater Monitoring Services

This parameter is about surface and underwater monitoring services, which are critical for understanding, protecting, and managing the complex system of water resources and marine ecosystems [184]. By collecting and analyzing data, these services provide valuable insights that can inform policies and practices to protect our natural resources and ensure their sustainable use for generations to come. These services rely on a variety of specialized equipment and techniques [185], including IoT, autonomous underwater vehicles (AUVs), and acoustic systems, to collect data on the water surface and underwater environments, including marine life, climate change, statistics of the inhabitants of coral reefs, water quality, temperature, salinity, and other physical and chemical properties. In rivers and oceans, acoustic sensors can be used to monitor the movement of marine life and the presence of underwater structures, such as oil rigs or offshore wind turbines. These data can be used to assess the impact of human activities on marine ecosystems and identify potential hazards. Additional examples of research related to this parameter include [186,187].

### 4.6.8. Precision Agriculture Services (PAS)

This parameter helped to cover multiple dimensions of precision agriculture services (PAS) such as productivity enhancement, resource management, precision prediction, smart irrigation systems, e-commerce for agricultural products, crop pest detection, digital twin in agriculture, and sustainable agriculture management [188,189]. PAS is an advanced agricultural management approach that utilizes technology and data to optimize crop production and reduce waste [190,191]. This technique is widely used in both the agricultural and forest industries, where it is vital to increase productivity while minimizing environmental impact. PAS incorporates various technologies, such as GPS, sensors, drones, and machine learning algorithms, to collect data on factors that influence crop growth, including soil moisture, nutrient levels, temperature, and weather patterns. By analyzing the data collected through these technologies [192,193], farmers can make informed decisions about when and where to apply fertilizers, water, and pesticides, among other inputs. This parameter has been the focus of a considerable number of research studies, some of which are [194,195].

### 4.6.9. Emergency & Disaster Response Services

Emergency and disaster response services are essential in responding to natural disasters and other events that can cause significant damage to property and threaten

human life [196]. Emergency response teams utilize advanced systems to collect data and assess the situation to help coordinate the response efforts effectively [197,198]. This parameter covered multiple dimensions of emergency and disaster response services, including disaster and agriculture sentinel applications, web tools for remote services, satellite earth observation (EO) of vegetation regions, web services for broadcasting alerts and early warnings, typhoon rainfall forecasts in real-time, decision-making support, better rescue plans, and aid for both individuals and disaster management authorities. For instance, Doxani et al. [199] highlighted the importance of disaster and agriculture sentinel applications to reduce and support decision making for disaster management.

### 4.6.10. Public–Private Partnership (PPP)

PPP is a collaborative approach that brings together the resources and expertise of the public and private sectors to jointly fund, design, build, operate, and/or maintain public infrastructure, facilities, or services [200]. PPPs typically involve a government entity seeking to provide a service or build a system but lacking the financial or technical resources to do so on its own. This partnership approach relies on stakeholder participation, research, data analysis, and management to drive the design and implementation process [201]. PPPs can take various forms [202], including build–operate–transfer (BOT) contracts, concession agreements, joint ventures, or service contracts, and can be used to fund a wide range of projects, from transportation infrastructure to social infrastructure (such as schools and hospitals) and information technology systems. Ultimately, PPPs offer an opportunity to leverage the strengths of both the public and private sectors to create sustainable and innovative solutions that benefit society as a whole. Additional examples of research related to this parameter include [203,204].

### 4.7. Digital Transformation

This category includes services that help organizations leverage technology to transform their operations and processes [205,206]. It focuses on using digital tools and technologies to improve efficiency, productivity, and innovation. The services in this category are geared towards organizations that want to stay competitive in a rapidly changing digital landscape. Some of the services in this category include satellite navigation services, indoor positioning services (IPS), online and social commerce services, edge computing services, IoT and network security services, and privacy protection services. These services are designed to help organizations make better use of data, improve their online presence, and secure their digital assets.

### 4.7.1. Satellite Navigation Services

Satellite navigation services, also known as global navigation satellite systems (GNSS), are a network of satellite-based positioning systems that provide location and time information to users around the world [207,208]. These services use a constellation of satellites in orbit around the Earth to transmit signals to a receiver on the ground, which determines its precise position. The most well-known system is GPS [209], and other systems include the Russian GLONASS [210], the European Galileo [211], the Chinese BeiDou (BD) [212], and more. These systems work by measuring the time it takes for signals to travel from the satellites to the receiver on the ground, enabling the receiver to determine its position with high accuracy. Satellite navigation services have a wide range of applications, including navigation for aircraft, ships, and vehicles, surveying and mapping, disaster prevention and management, including geolocation for emergency services, security management, communications, and tracking of people and objects. This parameter helped to cover the different dimensions of GNSS, including those mentioned above, such as precise point positioning, GNSS effectiveness and comparisons, vehicle tracking with GNSS, and GNSS applications in police and insurance. For instance, see [213,214].

### 4.7.2. Indoor Positioning Services

Indoor positioning services are becoming increasingly important as people spend more time indoors, especially in large commercial or public buildings [215,216]. IPS use a combination of technologies to accurately locate objects and people in indoor environments, making it an essential tool for navigation and wayfinding. IPS rely on various signals, including Wi-Fi [217,218], BLE [219], magnetic positioning, and visual positioning, to create a fingerprint of the environment and provide accurate location information. Among the key benefits of IPS is their high level of accuracy. With an accuracy of up to several centimeters, IPS can help people navigate indoor spaces more efficiently and reduce the time spent searching for a specific location [220].

In addition to navigation, IPS can also be used for asset tracking, which helps organizations optimize their operations and reduce costs. IPS can also provide valuable data insights into how people move through indoor spaces [221]. These data can be used to optimize building layouts, improve traffic flow, and identify areas for improvement. Moreover, in emergency situations such as fires, floods, or earthquakes, IPS can quickly locate people and assets, allowing for a timely and effective response. As buildings become more complex and users' expectations increase, IPS will continue to play an essential role in providing an accurate, efficient, and enjoyable indoor experience. This parameter covered a range of IPS dimensions, including those mentioned above, such as indoor localization, navigation and wayfinding, physical activity recognition, smart homes and public well-being, indoor/outdoor environment identification, and IoT localization. For example, see [222,223].

### 4.7.3. Online & Social Commerce Services

Online and social commerce services provide a powerful tool for businesses to grow their brand and increase revenue by creating a more customer-centric approach to commerce [224]. These services have transformed the way businesses operate in the digital age; our model has identified multiple dimensions that enhance the customer experience [225]. These dimensions include technology, marketing, sales, customer experience, and data analytics. With online commerce, businesses can reach a wider audience, increase brand awareness, and drive consumer engagement by promoting their products and services through digital channels. Social commerce takes this one step further, enabling businesses to leverage social media platforms to engage with customers, build relationships, and gain valuable reviews and feedback [226,227]. By optimizing the customer experience, businesses can provide a seamless and enjoyable purchasing journey that encourages repeat purchases, fosters brand loyalty, and ultimately drives sales [228]. Through the use of data analytics, businesses can also monitor and improve their online and social commerce performance by tracking website traffic, consumer behavior, and product reviews [229]. Other dimensions of this parameter that our tool helped to identify include promoting co-creation services [230], supporting collaborative consumption and sharing economies [231], and enhancing social commerce in communities [232].

### 4.7.4. Edge Computing Services

Edge computing services are becoming increasingly popular due to their ability to provide a distributed computing architecture that brings computing resources closer to the network edge [233]. This proximity allows edge computing services to improve their performance and reduce network latency, a common problem in cloud computing [234]. The various architectures that are used in edge computing services, such as mobile edge computing (MEC), cloud computing, and fog computing, offer unique features and benefits that can be leveraged in different use cases [235].

One significant advantage of edge computing services is that they reduce the workload of centralized cloud computing resources [236]. By processing data closer to the source, edge computing services can offload tasks from the cloud and reduce the amount of data that needs to be transferred over the network [234]. This reduction in data transfer also

leads to a reduction in the energy consumption required for data transfer. Edge computing services also enable the deployment of new services that were not possible before due to the limitations of cloud computing [237]. Additional examples include [238,239].

### 4.7.5. Internet of Things (IoT) & Network Security Services

The IoT is a rapidly growing network of physical devices that are connected to the internet, enabling them to collect and exchange data. However, with the increase in the number of IoT devices being deployed, there is a significant need for network security services to ensure that the data collected and exchanged remains secure [240]. This is because the devices can be vulnerable to a range of security threats, including malicious attacks and distributed denial of service attacks (DDoS) [241]. IoT network security services are multifaceted and involve several critical dimensions to ensure the security of IoT devices and the data they collect and exchange [242]. One of the key dimensions is the ability to detect and prevent attacks, which is crucial for maintaining the integrity and security of IoT networks [243]. Techniques such as firewalls and intrusion detection or prevention systems can be used to monitor network traffic to and from IoT devices and detect any potential threats [233]. Additional examples for this parameter include [244,245].

### 4.7.6. Privacy Protection Services

In today's digital age, the protection of privacy and data security has become a growing concern [246]. Personal data are being collected and stored online at an increasing rate, which makes it essential to ensure that this information remains secure and private. This is where privacy protection services come in [247]. Privacy protection services are designed to safeguard personal data and prevent unauthorized access, use, or disclosure [248]. Encryption is a key technology used in these services [249,250]. There are several other important technologies used in privacy protection services, such as IoT access control, privacy-aware authentication, single sign-on, virtual identity, and identity-as-a-service (IDaaS). For instance, Zhang et al. [251] presented a secure smart health (SSH) system with IoT access control and privacy-aware aggregate authentication.

### *4.8. Service Lifecycle Management*

This category includes services that help organizations manage the entire lifecycle of their products and services [252]. It covers everything from service modeling and pricing to quality and innovation. The services in this category are geared towards organizations that want to improve the customer experience, streamline their operations, and stay competitive. Some of the services in this category include service modeling, service pricing, service quality and experience, business innovation services, and Services 5.0. These services are designed to help organizations create new service offerings, improve existing ones, and better manage their customer relationships throughout the service lifecycle.

### 4.8.1. Service Modeling

Service modeling is a process that involves creating a detailed representation of a service system by analyzing its various components, relationships, and functionality [253]. Service modeling is an essential tool for service designers and managers [254], allowing them to gain a deep understanding of the complexities of service systems. By identifying potential inefficiencies and bottlenecks, service designers can make informed decisions and optimize the system to provide the best possible service. This optimization can result in increased customer satisfaction and loyalty, ultimately leading to higher revenues for the business. Additional examples include [255,256].

### 4.8.2. Service Pricing

The pricing of services is an essential aspect of our modern economy [257]. It is a critical aspect of any service-based business, as it determines the generated revenue and ultimately affects the profitability of the business [258]. Pricing a service requires the careful

consideration of various factors, including cost, competition, value proposition, market demand, profit margin, seasonal factors, and consumer satisfaction. Competition in the market plays a crucial role in determining the price of services [259,260]. Market demand can also affect pricing [261]. Profit margin is also a crucial factor to consider when pricing services [262].

However, it is essential to recognize that pricing strategies are not always fair to consumers [263]. In some cases, businesses use complex pricing models that take advantage of information asymmetry to charge higher prices than necessary. This can lead to market inefficiencies and unfairness for consumers. Furthermore, pricing strategies can also have significant impacts on the environment and society. For instance, some pricing models encourage excessive consumption and waste, leading to negative environmental impacts. Additional examples of research related to this parameter include [264,265].

### 4.8.3. Service Quality & Experience

This parameter covered multiple dimensions of service quality and experience that refer to the level of performance and satisfaction users receive from a product or service [266]. It is an essential aspect of any industry that provides a service, from healthcare to hospitality, from banking to technology. By providing high-quality, reliable, and efficient services, businesses can increase customer satisfaction, loyalty, and profitability. Service quality refers to the level of satisfaction that a customer experiences when interacting with a business or organization [267,268]. It is evaluated based on factors such as reliability, responsiveness, empathy, assurance, and tangibles. In healthcare, service quality and experience can be the difference between life and death. It is essential to provide a safe, timely, and effective service, along with empathy and professionalism, to ensure patients receive the best care possible [12]. In the hospitality industry, service quality and experience can impact customer loyalty and repeat business. By providing a positive service experience through a clean and welcoming environment, attentive and friendly staff, and high-quality amenities, hotels and restaurants can attract and retain customers [31]. A large proportion of articles in this parameter were focused on communication networks, such as [269,270].

### 4.8.4. Business Innovation Services

Business innovation services refer to a wide range of specialized solutions that help businesses identify, develop, and implement innovative ideas, products, services, and processes [271]. These services are essential for businesses to succeed in today's rapidly changing marketplace, enabling them to adapt and thrive in the face of constant disruption and uncertainty [272,273]. These services cover a range of areas, including ideation, technology, strategy, marketing, and training [274,275]. This parameter—i.e., business innovation services—covered multiple dimensions, including digital transformation [276], product development [277], process optimization, and service innovation [278]. Additional examples of related research can be found in [279,280].

### 4.8.5. Services 5.0

This parameter covered the different dimensions of Services 5.0, which is a digital, cloud-based, and smart service delivery model that leverages advanced technologies to provide personalized, proactive, and sustainable services [281]. Services 5.0 represents the future of service delivery, characterized by personalization, proactivity, co-creation, and sustainability [282]. This evolution is driven by advanced technologies such as IoT; cloud, fog, and edge computing; big data analytics; digital twins; and social media.

In Services 5.0, customers play an active role in the design and delivery of services through co-creation [283]. This is facilitated by social media and collaborative tools that allow customers to provide feedback and suggestions in real-time. Services 5.0 is characterized by a focus on sustainability [284], with service providers seeking to minimize their environmental impact and maximize their social impact. This is achieved with green technologies and sustainable business practices, as well as by engaging with customers

and other stakeholders to promote social responsibility. Service 5.0 is an advanced service delivery model that builds on the foundation of Service 4.0. While both models integrate advanced technologies such as IoT, cloud computing, and big data analytics, Service 5.0 adds new features, including personalization, proactivity, co-creation, and sustainability. Service 5.0 is more customer-centric and collaborative, with a focus on tailoring services to individual needs and preferences, anticipating customer needs, involving customers in service design, and promoting environmental and social responsibility.

## 5. Regional Analysis Results: Service Sector Parameters from Twitter

In this section, we describe the outcomes of our tool's analysis of the Arabic Twitter dataset concerning different services offered in Saudi Arabia. We provide a summary of the parameter space in Section 5.1. The modeling-related findings are quantitatively assessed in Section 5.2. In Sections 5.3 and 5.4, we examine each macro-parameter individually, along with its associated parameters, and furnish instances of Twitter posts to support our explanations.

### 5.1. Parameters and Macro-Parameters: An Overview

After analyzing the Twitter dataset related to the services sector, we identified a total of 15 clusters. To avoid redundancy, we merged some clusters with similar themes, which led to the identification of 11 parameters, organized into 2 macro-parameters, namely private sector services and government services. Table 2 presents the results in the same format as Table 1 (see Section 4). Figure 5 gives a taxonomy of the parameter space of the Twitter dataset.

**Table 2.** The service sector parameters (source: Twitter).

| Macro-Parameters | Parameters | Cluster No. | % | Keywords |
|---|---|---|---|---|
| Private Sector Services | Food Delivery Apps & Services | 1 | 5.47 | Hungerstation App, Jahez App, Mrsool App, Jeeny App, Registration, ToYou App, Driver, Benefit, Applications, Shgardi App, Applications, Code, Inquiry, Order It, Healthy |
| | | 14 | 2.28 | Mrsool App, Hungerstation App, Jahez App, To You App, Registration, Apps, Inquiry, Whatsapp, Apps, Free, Coupon, Shgardi App, To Order, By Whatsapp, Quality |
| | Logistics Services | 7 | 3.93 | Naqel, Car, Zajil, Transport, Order, Prices, SMSA, Check, Payment, Sale, The Naqel, Possibility, Use, Transfer, Snapchat |
| | Ride-Hailing Services | 0 | 5.77 | Uber, Registration, Jeeny, Rides, Taxi, Train, Shopping, Code, Discount, Trip, Coupon, Careem, Applications, Traveler, Transportation |
| | | 12 | 3.05 | Uber, Mobily, Trip, Man, Transportation, Rides, Able, Taxi, Easy, Pictures, Drivers, Needy, Metro, Stop, The Only One |
| | | 13 | 2.43 | Uber, Careem, Airport, Cheapest, Metro, Naqel, Trip, Cars, Healthy, General, Trip, Transport, Driver, Passenger, Car |
| | Mobile & Broadband Services | 3 | 5.20 | Mobily, Zain, Service, Customers, Installment, Online, Easily, Immediately, Shopping, Payment, Package, Services, Bill, Viber, WhatsApp |
| | | 10 | 3.11 | Mobily, Store, Electronic, To Stop, Send, Its Features, Qitaf, Purchase, Smart, Its Offers, Available, Messages, Devices, Packages, iPhone |
| Government Services | Passport & ID Services | 2 | 5.30 | Appointment, Reservation, AljawazatKSA, Requires, Passport, Receipt, Travel, Procedure, Electronic, E-Mail, My Health, To Check, The Sessions, The Session, The Kindergarten, The Judiciary |
| | Digital Identity Services | 6 | 4.10 | Mobily, Systems, Access, Unified, Identity, Postpaid, Shop, To Mobily, Bills, Electronic, The Offer, Offer, Ease, Transfer, Payment |
| | Police Services | 11 | 3.06 | Communication, Kollona Amn, Najiz, Procedure, Number, We Wish, Requests, Mobily, Request, Through, Send, Our Client, Log In, To Help You, Corruption |

**Table 2.** *Cont.*

| Macro-Parameters | Parameters | Cluster No. | % | Keywords |
|---|---|---|---|---|
| Government Services | Judiciary Services | 8 | 3.85 | Najiz, Judiciary, Lawsuit, Agency, Issuance, Entry, Court, Inquiry, Agency, Service, Requests, Platform, Required, Justice, Steps |
| | Social Security Services | 4 | 5.09 | Zajil, Sakani, The Citizen's Account, Hafez, Download, Social Security, Electronic, Developer, Appointments, Social, The Reservation, Reservation, Document, Problems, Financing |
| | Workforce Development Services | 5 | 4.84 | Communication, Text, Government, Messages, Administrative, Announce, Job, Technical, Application, Jobs, Government, Technology, Fields, Employment, Through |
| | Digital Transformation Services | 9 | 3.83 | Services, Economy, Transportation, Communications, Judiciary, Growth, Judgments, Judicial, Department, SAR (Saudi Arabia Railways), Digital, Orders, Najiz, Smart, Entry |

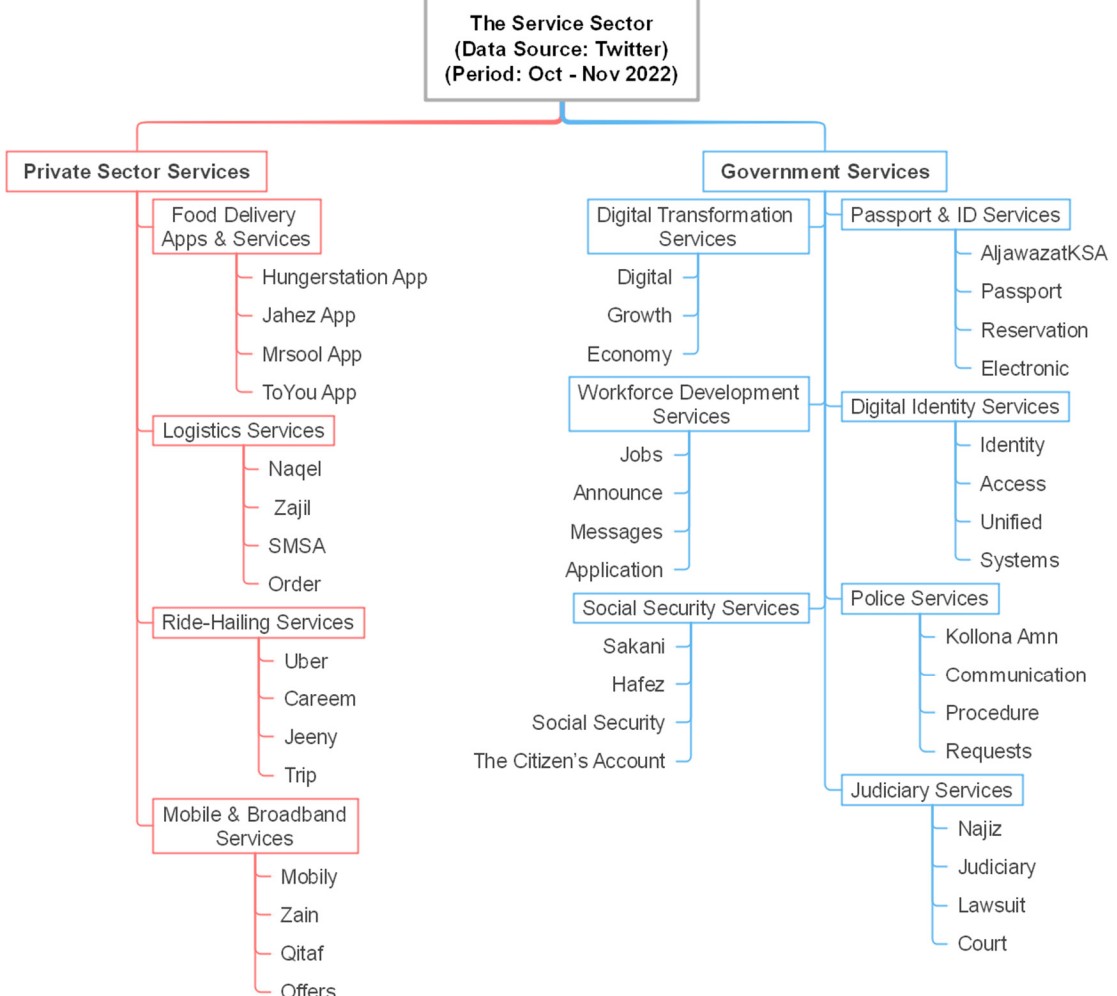

**Figure 5.** Taxonomy of the service sector from the Twitter dataset.

*5.2. Quantitative Analysis (Twitter Dataset)*

Similar to Section 4.2, we conducted a quantitative analysis of the results obtained with modeling parameters using our software tool and the Twitter dataset. Our analysis includes a term score decline graph, an inter-topic distance map, a similarity matrix, keyword scores, and hierarchical clustering.

The "term score decline" results indicate that the difference between keyword scores for clusters becomes insignificant after 13 to 17 terms. This suggests that having 13 to

17 terms in the ranking of each topic adequately captures the concept of the topic. This information was used to examine the top 15 related parameter keywords. The c-TF-IDF score was used to rank the parameter keywords (see Section 3.5).

The distance map between topics, the hierarchical clustering map, and the similarity matrix, along with domain expertise, were employed to develop parameters and group them into macro-parameters. We did not include these figures in the paper to avoid excessive length. We would be happy to provide these graphs upon the reader's request.

### 5.3. Private Sector Services

This macro-parameter gathers four parameters that relate to the services offered by the private sector. Private sector services are provided by private individuals or organizations to meet consumer needs and generate profits [285]. They range from basic necessities to specialized offerings across sectors such as finance, healthcare, education, retail, hospitality, and technology. Private sector services play a significant role in the economy, contributing to employment, GDP, and meeting consumer demands.

#### 5.3.1. Food Delivery Apps & Services

Food delivery apps and services are online platforms that allow users to order a wide range of cuisine options from multiple restaurants, customize their orders, save time, ensure safety, and receive loyalty benefits [286,287]. Popular services include Uber Eats, Grubhub, DoorDash, Postmates, and Deliveroo. Our tool detected from the tweets that a number of food delivery apps were being used in Saudi Arabia, including Hungerstation, Jahez, Mrsool, Jeeny, ToYou, and Shgardi. The following tweet exemplifies this parameter: "We do not provide food delivery services, but it is possible to order from mobile apps such as Mrsool, Jahez, Hungerstation, or ToYou".

#### 5.3.2. Logistics Services

Logistics services involve the planning, implementation, and control of the movement of goods and services [288,289]. This includes transportation, warehousing, inventory management, and information flow management [61]. The goal is to ensure that products are delivered efficiently, on time, and at the lowest possible cost. With the increasing trend of people buying online, the burden on shipping companies has increased, and competition has flared up among them to provide the best services at competitive prices, with the importance of maintaining the security of transported goods and their delivery without damage. The following tweet exemplifies this parameter: "I have dealt with all the shipping companies including SMSA, Aramex, and Naqel, and have found that they all deliver orders as quickly as is possible".

#### 5.3.3. Ride-Hailing Services

Ride-hailing services provide on-demand transportation through a mobile application [290], connecting passengers with nearby drivers who use their personal vehicles. These services have gained popularity due to their convenience and low cost, and major companies include Uber, Lyft, Grab, and DiDi. Additionally, these services improve mobility for low-income and elderly people, as well as disabled persons [291]. Our tool has identified these services in Saudi Arabia from various companies, including Uber, Jeeny, and Careem. In Saudi Arabia, people are also using these services for other tasks, such as calling a taxi and requesting to bring food or perform other chores, as expressed in the following tweet: "I prefer using Careem and Uber for my deliveries because they are cost-effective. Typically, the price of delivery remains within 25 Saudi Riyals".

### 5.3.4. Mobile & Broadband Services

Mobile services use mobile networks to allow communication using portable devices [292,293], while broadband services provide high-speed internet access through various technologies such as cable, fiber, DSL, and satellite. They enable a range of applications, including video conferencing, online gaming, cloud computing, and remote work, and are essential for individuals and businesses to stay connected and access information anytime and anywhere. Our tool has identified these services in Saudi Arabia from various companies, such as STC, Mobily, Zain, and the Qitaf service.

### *5.4. Government Services*

This macro-parameter gathers seven parameters that relate to the services offered by the government sector. Government services are the programs and initiatives provided by the government to its citizens [294,295]. These services are developed to meet the requirements of the people, to promote their welfare, and to ensure their safety and security. The government's services may vary depending on the country and its political system. Examples of government services include passport & ID services, digital identity services, police services, judiciary services, social security services, workforce development services, and digital transformation services. These services provide assistance with travel documents, online identity verification, maintaining law and order, administering justice, providing financial assistance, improving employability, and promoting transparency and efficiency in government operations.

### 5.4.1. Passport & ID Services

Passport and ID services are government services that provide citizens and residents with identification documents [296], including passports, national IDs, and driver's licenses. These documents are essential for verifying identity and citizenship, facilitating travel, and accessing government services and benefits. In Saudi Arabia, these services indicate, for instance, that government documents can be created and renewed electronically—such as passports—through the Absher platform, which saves time and effort, as mentioned in the following tweet: "The Civil Status and Passports Department has introduced new electronic services for issuing family books".

### 5.4.2. Digital Identity Services

Digital identity services enable the secure and convenient establishment and verification of online identities through authentication and verification services, biometric identity services, and identity management platforms [297,298]. These services play a crucial role in ensuring secure online transactions and communication while protecting users from identity theft. An important example of digital identity services that our tool detected from Twitter data is the Nafath service in Saudi Arabia. This is because a number of government and private organizations have begun offering services through electronic portals and electronic identifiers (using usernames and passwords) as a verification tool that allows entry to their electronic service sites in Saudi Arabia. The Ministry of Interior has launched a national initiative to produce and control digital identities for citizens and others. It is referred to as a "digital identity" and is related to individuals, serving as a representation of their identities in electronic transactions. This initiative aims to find a comprehensive solution for managing digital identities to enable unified electronic access services at the national level, enabling governments and private sector organizations to contribute to the delivery of different services. The following tweet highlights this: "Dear customer, verify your account through the unified national access (Nafath) to enjoy all STC Pay services easily in few steps".

### 5.4.3. Police Services

Police services are a critical component of law enforcement [299,300], responsible for maintaining public safety, enforcing the law, and preventing crimes. They work closely with other law enforcement agencies and community organizations to promote public safety. The effectiveness of police services is measured by their ability to prevent and solve crimes and maintain good relationships with the communities they serve. Kollona Amn, detected with our tool, is a service that enables Saudi Arabian citizens and residents to assume the position of a police officer. This expedites rescue operations and lessens losses and damages. Residents and citizens can send an incident with a video, photo, or voice remark attached. The following tweet highlights this: "Electronic crimes can be reported through the Kollona Amn app".

### 5.4.4. Judiciary Services

Judiciary services are responsible for interpreting and enforcing laws, resolving disputes, and ensuring justice is served [301,302]. The branch includes judges, lawyers, and court personnel who work together to maintain the balance of power between branches of government and uphold the rule of law. In Saudi Arabia, the government is keen to unify judicial procedures and the completion of legal transactions electronically. Najiz is an electronic platform affiliated with the Saudi Ministry of Justice that aims to facilitate court services provided to beneficiaries, including citizens, residents, and business owners, and complete the beneficiaries' transactions in agencies, courts, real estate, and others. This is highlighted in the following tweet: "Najiz is a judicial system that offers all the services of the Ministry of Justice in the areas of judiciary, implementation, and documentation through a team of specialized legal professionals".

### 5.4.5. Social Security Services

Social security services are an important aspect of many countries' social welfare systems [303,304], offering financial support to individuals and families in the event of retirement, disability, or death. These programs are administered by government agencies and may include retirement benefits, disability benefits, survivor benefits, income supplements, and healthcare coverage. The specific types of programs and eligibility requirements may vary by country. This parameter encompasses the range of social security services introduced by the Saudi government to sustain communities and meet their needs. These services include financial, family-related, healthcare, housing, employment, psychological care, agricultural, and others. Examples of these services are the Sakani, Reef, and Hafez programs.

### 5.4.6. Workforce Development Services

Workforce development services are programs and initiatives designed to support individuals in developing the skills, knowledge, and experience needed to enter, re-enter, or advance in the workforce [305,306]. These services can be offered by government agencies, non-profit organizations, or private companies and may include job training and skills development, education and credentialing, career counseling and guidance, and job placement and support. The overarching goal of these services is to help individuals achieve their career goals and succeed in the workforce. An example of the tweets in this parameter is as follows: "The unified national platform for employment simplifies various stages for job applicants, whether in the private or government sectors, and saves time in securing the right job".

### 5.4.7. Digital Transformation Services

Digital transformation services are a set of strategies, processes, and technologies that help businesses, governments, or other organizations leverage digital technologies to enhance their operations, develop novel ways of conducting business, and enhance customer satisfaction [205,307]. It involves developing a comprehensive digital strategy,

implementing the right digital technologies, managing data and analytics, change management, and digital marketing. The digital transformation of services has led to the prosperity of countries, the expansion of activities, and the growth of economies around the world. This parameter helped to cover digital transformation activities in Saudi Arabia, as exemplified with the following tweets: "Government agencies are making great progress in their digital transformation journey. The Digital Government Authority has now announced the government agencies that have reached the stage of creativity and are the most advanced in measuring digital transformation in 2022" and "At the Digital Government meeting, the Ministry of Justice was ranked first in terms of digital transformation among government agencies".

## 6. Synthesis: Global vs. Regional Analyses and Research Implications

This research aims to develop a methodology that employs ML to gain a thorough understanding of the service sector, with the ultimate goal of creating a theory and approach for smarter services and service economies that support sustainable future societies. The study aims to drive future research in this field and achieve this objective through the development of autonomous systems using innovative technologies and solutions.

Earlier, in Figure 1, we depicted a multi-dimensional taxonomy that offers a comprehensive overview of the service sector, combining multiple perspectives from academia and the public. The taxonomy presents a global outlook and a localized perspective specific to Saudi Arabia, emphasizing the richness of the sector. The global and academic perspective covers a wide range of services, such as education, healthcare, transportation, smart society and infrastructure, digital transformation, and service lifecycle management. It provides insights into the use of technology to improve service delivery and develop new business models, and it offers a comprehensive understanding of the current trends and developments driving the sector's evolution. The local and public view provides insights into the types of services offered by the private and government sectors in Saudi Arabia, helping to identify gaps and develop strategies to improve service delivery.

The identification of these macro-parameters and their associated services highlights the depth and breadth of the service sector and provides valuable insights into the various areas of service provision. By utilizing innovative technologies and service offerings, organizations can improve the customer experience, streamline their operations, and stay competitive in today's rapidly changing digital landscape. In Figure 2 (Section 1), we provide a framework for autonomous design and operations of the service sector, which is developed using the parameters and information discovered using AI in this work. The process of developing such frameworks can be automated using our approach in any field.

The study used a data-driven methodology to model the service sector by combining the academic literature and public opinion gathered from Twitter. ML and big data are leveraged to extract key parameters from both sources to gain a unique understanding of the service sector from two distinct perspectives. The study developed a software tool for automated analysis and capture of parameters. The data availability or data-drive/data-centric nature of our methodology is integral to the methodology of AI-based solutions. It enables model training, generalization, and fairness, reducing bias and enhancing robustness. Access to diverse and sufficient data ensures AI systems perform accurately, ethically, and reliably, making it a foundational element of AI development.

Analyzing the Twitter data from a particular country about service economies can provide valuable insights into trends and patterns, customer sentiment, the impact of government policies, and new market opportunities. This information can be used by businesses to develop new products and services, improve their existing offerings, make better strategic decisions, and improve customer service. Governments can use Twitter data to develop more effective policies, and researchers can gain new insights into the service economy.

Analyzing the Twitter data from Saudi Arabia about service economies can be especially beneficial given the large and growing Saudi Arabian service sector, the popularity

of Twitter in Saudi Arabia, and the wide range of insights that Twitter data can provide. Specific examples of how Twitter data have been used to analyze the service economy in Saudi Arabia include tracking the growth of the e-commerce industry, measuring the impact of the COVID-19 pandemic, and identifying the most popular services in different Saudi Arabian cities. Overall, analyzing the Twitter data from Saudi Arabia about service economies can be a very valuable tool for businesses, governments, and researchers.

Combining insights from global academic analyses with regional Twitter data provides a holistic view of the service sector. The global analysis highlights industry trends and technological advancements, while the regional perspective, focused on Saudi Arabia, reveals specific service offerings and opportunities for improvement. This synthesis enables more informed decisions and strategies that align local practices with global trends.

### 6.1. Theoretical and Practical Implications

The work holds substantial consequences both in theory and application, not only for the service industry but also for other fields. In today's fast-paced and ever-changing technological environment, this study is of the utmost importance.

The service industry is continuously facing numerous challenges, including globalization, geopolitical risks, conflicts, economic downturns, technological changes, climate change, demographic changes, income disparities, and regulatory reforms. Previous research (Section 4) and related works (Section 2) indicate that the service sector has a narrow and fragmented focus. To tackle these challenges effectively, it is essential to adopt a holistic approach to studying the service sector, considering economic, social, environmental, governance, and cultural factors. This comprehensive view provides decision makers with a deeper understanding of the interconnectedness of various aspects of the service industry and their impact on each other. It also helps identify potential challenges and opportunities, enabling proactive solutions that stay ahead of economic trends and mitigate negative impacts. A holistic approach leads to sustainable economic and service practices, promoting better quality service and experience for all, benefiting societies, economies, and the planet.

In the current digital era, the impact of service economies and the service sector on local cultures, particularly in Saudi Arabia, has not been extensively researched. Specifically, the analysis of big data from social and digital media in Arabic-language contexts is still scarce. Saudi Arabia's Vision 2030 plan outlines a significant transformation in the country's economic and social structure, with a goal of expanding beyond the oil industry and creating a renewed social contract. The plan envisions a thriving economy achieved through a new economic model that prioritizes increased productivity, with the government shifting from an economic driver to an enabler of private-sector-led growth. The plan focuses on improving human capital, attracting foreign investment through an enhanced business environment, establishing a high-quality public administration, and creating a flexible and competitive labor market [308–310]. The services sector is a main driver of economic growth in Saudi Arabia, as the government has prioritized its development and investment. Research in this sector can provide insights into opportunities and challenges facing the sector, the effectiveness of policies, competitiveness compared to other countries, and barriers to entry for new businesses. Such research can be valuable for policymakers, businesses, and investors seeking to capitalize on the growth potential of the services sector [311,312].

The study presents a new approach for gathering and analyzing diverse and impartial information related to the service industry using advanced ML techniques. The method allows for the examination of multiple datasets to uncover valuable insights that can benefit different stakeholders, such as academic researchers, corporate decision makers, government officials, and the public. The study's findings contribute significantly to the understanding of the service industry and can provide a foundation for future research and informed decision making.

Moreover, using the discovered knowledge in this work, a framework (Figure 2) for autonomous design and operations of the service sector is developed in this paper using data analytics, AI, machine and deep learning, and robotic systems to improve service quality and efficiency, reduce costs, and enhance customer satisfaction while ensuring data security and compliance with ethical and legal considerations. The framework involves identifying services to be automated, developing architectures, choosing the right technology, implementing cybersecurity measures, establishing monitoring and feedback mechanisms, addressing ethical and legal concerns, and developing a roadmap for implementation.

This study proposes a new method for utilizing AI techniques to gather and analyze comprehensive and adaptable information from various sources, such as social media, the scientific literature, and other datasets. This approach can be applied to optimize systems and applications and facilitate autonomous capabilities for information discovery and parameter identification. Moreover, it has the potential to promote innovative research in the service sector and contribute to the creation of sustainable, responsible, and smarter economies. As the trend towards autonomy in various systems continues, understanding the system parameters is becoming increasingly important for effective problem solving and decision making.

*6.2. Contributions to Sustainability and United Nations Sustainable Development Goals (SDGs)*

The paper aligns its findings and approach with the UN SDGs and relevant sustainability frameworks, displaying a dedicated commitment to global sustainability goals. Regarding SDGs, it emphasizes technology-driven optimization and innovation in line with SDG 9 (Industry, Innovation, and Infrastructure) for sustainable industrialization. The focus on creating smarter and more sustainable service economies resonates with SDG 11 (Sustainable Cities and Communities), promoting inclusive, safe, and resilient urban environments. Examining sustainability in service delivery contributes to SDG 12 (Responsible Consumption and Production), reducing waste and resource use. Addressing environmental concerns aligns with SDG 13 (Climate Action).

In terms of other sustainability frameworks, the paper's autonomous service design embodies Circular Economy Principles, enhancing circularity. It considers social and environmental dimensions, reflecting Triple Bottom Line (TBL) sustainability. The local service economy study (Section 5), as well as other relevant findings from Section 4, align with the Local Sustainability Initiatives addressing region-specific challenges.

The paper champions a Holistic Sustainability Approach, acknowledging interconnectedness among economic, social, cultural, and environmental dimensions. It advocates a holistic approach to service economies, underscoring the need to integrate sustainability from the outset. Proactive planning is imperative for enhancing reputation, supporting viability, fostering societal impacts, fueling innovation, and minimizing environmental footprints.

In summary, the paper's significance lies in its dynamic comprehension, cultural sensitivity, and holistic approaches to addressing global challenges and advocating smarter, sustainable service practices. Its multifaceted perspective and advanced technologies, including big data and machine learning, position it as a valuable asset in the pursuit of sustainability in service economies, contributing significantly to the literature.

## 7. Conclusions and Outlook

This study presents a data-centric approach to modeling the service sector, using the academic literature and public opinion from X (formerly known as Twitter). Advanced technologies were used to extract key parameters from both sources, and a software tool was developed for this purpose. After analysis, 29 distinct parameters related to the service sector were identified from research articles, while 11 parameters related to public opinion on the service sector were identified from tweets collected in Saudi Arabia. The software tool was used to generate a knowledge structure, taxonomy, and framework for the service sector, in addition to a literature review based on over 300 research articles. In

conclusion, as highlighted in the previous section, this study has noteworthy practical and theoretical implications for the development of autonomous capabilities in systems, which can contribute to the creation of smarter and more sustainable societies.

It is important to consider the limitations of this paper's findings. One limitation is the narrow focus on the service sector in Saudi Arabia, which may not be applicable to other regions or contexts. Another limitation is the reliance on data sources for the proposed approach that utilize ML, as the accuracy and reliability of these sources can impact the quality of the results. Furthermore, the complexity of the algorithms employed can make interpretation difficult and potentially introduce biases. These two limitations can be addressed by conducting similar studies on social media data from different countries and the academic literature data from different databases. Lastly, the approach relies heavily on technological infrastructure, which may not be available in all contexts, potentially limiting its applicability to decision makers in certain areas. This can be addressed by providing digital capabilities to those areas or by carrying out similar analyses for those areas and providing them with the insights for decision making.

This paper is part of our ongoing research into using ICT to address challenges in digital societies. The research has covered a wide range of topics, including education [80], healthcare [109], smart families [49], tourism and transportation [31,48], labor economics [35], and journalism [35], among others. Our aim is to enhance the methodology proposed in this article by utilizing cutting-edge machine learning techniques to tackle various challenges associated with the service sector and other social, economic, environmental, and cultural matters. To achieve this, we used data from the Scopus database and Twitter. Going forward, we intend to extend our research by incorporating more scientific databases, digital and social media, and other data sources to gain a more holistic understanding of these challenges.

We believe this study is an important step towards the formation of a theory and approach for smarter and more sustainable services and service economies that can support sustainable future societies through the development of autonomous systems using innovative technologies and solutions.

**Author Contributions:** Conceptualization, N.A. and R.M.; methodology, N.A. and R.M.; software, N.A.; validation, N.A. and R.M.; formal analysis, N.A., R.M., A.A., T.Y., and J.M.C.; investigation, N.A., R.M., A.A., T.Y., and J.M.C.; resources, R.M. and A.A.; data curation, N.A.; writing—original draft preparation, N.A. and R.M.; writing—review and editing, R.M., A.A., T.Y. and J.M.C.; visualization, N.A.; supervision, R.M. and A.A.; project administration, R.M.; funding acquisition, R.M. All authors have read and agreed to the published version of the manuscript.

**Funding:** The authors acknowledge, with thanks, the technical and financial support from the Deanship of Scientific Research (DSR) at the King Abdulaziz University (KAU), Jeddah, Saudi Arabia, under Grant No. RG-11-611-38.

**Institutional Review Board Statement:** Not applicable.

**Informed Consent Statement:** Not applicable.

**Data Availability Statement:** The data used in this paper can be made available subject to the data providers' terms and conditions.

**Acknowledgments:** The work carried out in this paper is supported by the HPC Center at the King Abdulaziz University. The training and software development work reported in this paper was carried out on the Aziz supercomputer.

**Conflicts of Interest:** The authors declare no conflict of interest.

## List of Abbreviations

| Acronym | | Acronym | |
|---|---|---|---|
| AI | Artificial Intelligence | ML | Machine Learning |
| BDS | Beidou Navigation Satellite | NLP | Natural Language Processing |
| BERT | Bidirectional Encoder Representations from Transformers | NLTK | Natural Language Toolkit |
| BLE | Bluetooth Low Energy | NSS | Navigation Satellite System |
| CRM | Customer Relationship Management | O2O | Online-to-Offline |
| D2D | Device-To-Device | OCC | Out-of-Class Communication |
| DDoS | Distributed Denial-of-Service Attacks | OGD | Open Government Data |
| DGs | Dangerous Goods | OI | Open Innovation |
| DOT | Department of Transportation | OSN | Online Social Networks |
| EAIT | Education and Information Technologies | PPP | Public–Private Partnership |
| EO | Earth Observation | PPP-AR | Precise Point Positioning with Ambiguity Resolution |
| EoS | Experience of Service | QoE | Quality of Experience |
| FUT | Fixed-Up-To | QoS | Quality of Service |
| G-D logic | Good Dominant Logic | REP | Retail Electric Providers |
| GNSS | Global Navigation Satellite System | RPAS | Remotely Piloted Aircraft System |
| GPS | Global Positioning System | SAML | Security Assertion Markup Language |
| GST | Goods and Services Tax | SAR | Saudi Arabia Railway |
| HDBSCAN | Hierarchical Density-Based Spatial Clustering of Applications with Noise | SDG | Sustainable Development Goals |
| ICTs | Information and Communication Technologies | S-D Logic | Service-Dominant Logic |
| IDaaS | Identity-as-a-Service | SDN | Software-Defined Network |
| IoT | Internet of Things | SIUS | Sociocultural Information Urban Space |
| IOV | The Internet of Vehicles | SM | Social Media |
| IT | Information Technology | SMA | Social Media Analysis |
| JIT | Just-In-Time | SSH | Secure Smart Health |
| JSON | JavaScript Object Notation | TBL | Triple Bottom Line |
| LDA | Latent Dirichlet Allocation | UAS | Unmanned Aircraft System |
| MAR | Mobile Augmented Reality | UAVs | Unmanned Aerial Vehicles |
| MEC | Mobile Edge Computing | UMAP | Uniform Manifold Approximation and Projection |
| MidSiot | Multistage Intrusion Detection System for Internet of Things | VID | Virtual Identity |
| MANETs | Mobile Ad Hoc Networks | VSEC | Vehicular Social Edge Computing |

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
