# Peer review of "Autonomous and Sustainable Service Economies: Data-Driven Optimization of Design and Operations through Discovery of Multi-Perspective Parameters"

_sustainability, doi:10.3390/su152216003_

Round 1

Reviewer 1 Report

Comments and Suggestions for Authors

the big plus of text is the huge sample and innovative method

but the text is definitely too long and discusses too many issues

I am suggesting splitting it into two coherent papers

citation in the first part is too broad - it is better to pick papers that explicitly refer to your research

conclusions are not supported by the results of your board research

the methodological part covers too many aspects which brings chaos with interpretation and linking it all together as a coherent paper

I am suggesting changing the subtitle to "Literature review"

Reviewer 2 Report

Comments and Suggestions for Authors

From my point of view the statistical framework is fine and accurate. However, probably the figures and the details could be integrated with annex, since the report is long, and it needs less pages and more accurate to understand the target of the paper. It is originally and could be a good report.

Reviewer 3 Report

Comments and Suggestions for Authors

Dear author

While this manuscript has chosen an interesting topic and is fairly well organized, it is imperative that the manuscript moves out of the thesis mode and is presented in the format of a valid scientific paper.

It is necessary to reduce the pages significantly to make it easier for the audience to read.

The basic findings of the research should be presented and additional items should be removed.

Good luck

Reviewer 4 Report

Comments and Suggestions for Authors

Thank you for the opportunity to review the manuscript titled "Autonomous and Sustainable Service Economies: Data-Driven Optimization of Design and Operations Through Discovery of Multi-Perspective Parameters." The research topic is timely and relevant, and the approach is innovative. Below are some observations and suggestions that might help enhance the clarity and impact of the manuscript.

Abstract: For the development of an article, it is ideal that its main goal represents the most important action the research aims to achieve. In its abstract, it is not clear what this goal would be.

The part "To this end, this study presents a data-centric method for modelling the service sector, using academic literature and public opinion from X (formerly known as Twitter)" might leave the reader confused, as it mentions the word "aim" without previously stating what it is, and then introduces the word "method". Speaking of "aim" alongside "method" can be scientifically risky. To do justice to this high-level article, I strongly recommend separating the objective into one sentence and the method into another.

For your objective, I suggest you make a mention, something along the lines of: The objective of this study is to develop and validate an artificial intelligence-based methodology to gain a comprehensive understanding of the service sector, identifying key parameters from academic literature and public opinion.

This methodology aims to provide in-depth insights into the creation of smarter and more sustainable services and service economies, contributing to the development of sustainable future societies.

I noticed that the study adopts a global scope for the analysis of academic articles and a regional scope (Saudi Arabia) for the analysis of Twitter posts. I believe this methodological choice warrants a more detailed explanation to provide clarity to readers:

Rationale for the Regional Scope of Twitter: It would be helpful to clarify why Saudi Arabia was specifically chosen for the Twitter analysis. Is there any particular relevance of Saudi Arabia to the service sector that makes its Twitter data especially pertinent to this study?

Relationship between Global and Regional Analyses: It would be beneficial to discuss how the insights derived from a global analysis of academic articles can be compared or contrasted with the insights from a regional analysis of Twitter posts. What is the value of combining these two distinct perspectives?

Clarification of Adopted Parameters: If there are specific methodological or practical reasons for the choice of these scopes (such as data availability, regional relevance, among others), this should be clearly indicated in the manuscript. I recommend that the authors address these points in the manuscript to strengthen the methodological foundation of the study and ensure readers have a clear understanding of the approaches adopted.

Discussion section: I believe it would be important to insert references in some passages to provide a foundation for the observations that were made. However, I believe that the discussions are taking place throughout the presentation of the results. Therefore, I think there would be the possibility for you to eliminate the discussion section and present the results section as "Results and Discussion".

Overall, the manuscript offers valuable insights into the service sector using a data-driven approach. The suggestions provided aim to enhance the clarity and structure of the paper, ensuring that readers can fully grasp the study's objectives, methods, and implications. I look forward to seeing the revised version of this manuscript and believe that with the suggested modifications, it can make a significant contribution to the field. This framing provides a positive and encouraging tone, emphasizing the value of the manuscript while offering constructive feedback for improvement.

Comments on the Quality of English Language

Dear Authors,

I have reviewed your manuscript titled "Autonomous and Sustainable Service Economies: Data-Driven Optimization of Design and Operations Through Discovery of Multi-Perspective Parameters." While the content and research approach are commendable, I noticed some areas where the English language could benefit from further refinement.

Ensuring that the manuscript is written in clear and precise English is crucial for conveying your findings effectively to the readers and the broader academic community. I recommend considering a thorough review or editing by a native English speaker or a professional language editing service. This will not only enhance the readability of your paper but also ensure that the nuances and intricacies of your research are accurately represented.

Thank you for your contribution to the field, and I look forward to seeing the revised version of your manuscript.

Best regards,

Round 2

Reviewer 4 Report

Comments and Suggestions for Authors

Article approved for publication. Congratulation!

Author Response

Thank you for recommending the manuscript for publication.